# MRAG-BENCH: VISION-CENTRIC EVALUATION FOR RETRIEVAL-AUGMENTED MULTIMODAL MODELS

**Wenbo Hu[1], Jia-Chen Gu[1], Zi-Yi Dou[1], Mohsen Fayyaz[1], Pan Lu[2],**
**Kai-Wei Chang[1], Nanyun Peng[1]**
[1]UCLA, [2]Stanford University
{wenbohu, gujc, zdou}@ucla.edu

https://mragbench.github.io

## ABSTRACT

Existing multimodal retrieval benchmarks primarily focus on evaluating whether models can retrieve and utilize external *textual knowledge* for question answering. However, there are scenarios where retrieving visual information is either more beneficial or easier to access than textual data. In this paper, we introduce a **m**ultimodal **r**etrieval-**a**ugmented **g**eneration benchmark, MRAG-BENCH, in which we systematically identify and categorize scenarios where visually augmented knowledge is better than textual knowledge, for instance, more images from varying viewpoints. MRAG-BENCH consists of 16,130 images and 1,353 human-annotated multiple-choice questions across 9 distinct scenarios. With MRAG-BENCH, we conduct an evaluation of 10 open-source and 4 proprietary large vision-language models (LVLMs). Our results show that all LVLMs exhibit greater improvements when augmented with images compared to textual knowledge, confirming that MRAG-BENCH is vision-centric. Additionally, we conduct extensive analysis with MRAG-BENCH, which offers valuable insights into retrieval-augmented LVLMs. Notably, the top-performing model, GPT-4o, faces challenges in effectively leveraging retrieved knowledge, achieving only a 5.82% improvement with ground-truth information, in contrast to a 33.16% improvement observed in human participants. These findings highlight the importance of MRAG-BENCH in encouraging the community to enhance LVLMs' ability to utilize retrieved visual knowledge more effectively.

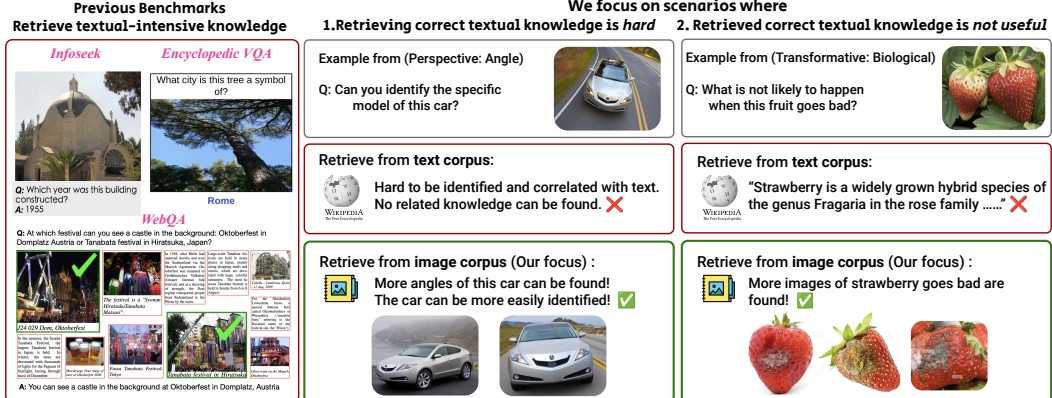

Figure 1: Example scenarios from MRAG-BENCH. Previous benchmarks (Chang et al., 2022; Mensink et al., 2023; Chen et al., 2023b) mainly focused on retrieving from textual knowledge. However, there are scenarios where retrieving correct textual knowledge is hard and sometimes not as useful as visual knowledge.

| Benchmarks | Knowledge Modality | Knowledge Source | Multi-Image Input | Diverse Scenarios |
|---|---|---|---|---|
| K-VQA (Shah et al., 2019) | Text | Wikipedia | ✗ | ✗ |
| OK-VQA (Marino et al., 2019) | Text | Wikipedia | ✗ | ✗ |
| MultiModalQA (Talmor et al., 2021) | Text | Wikipedia | ✗ | ✗ |
| ManyModalQA (Hannan et al., 2020) | Text | Wikipedia | ✗ | ✓ |
| A-OKVQA (Schwenk et al., 2022) | Text | Common/World | ✗ | ✗ |
| ViQuAE (Lerner et al., 2022) | Text | Wikipedia | ✗ | ✗ |
| WebQA (Chang et al., 2022) | Text/Caption | Wikipedia | ✗ | ✗ |
| Encyclopedia VQA (Mensink et al., 2023) | Text | Wikipedia | ✗ | ✗ |
| InfoSeek (Chen et al., 2023b) | Text | Wikipedia | ✗ | ✗ |
| MRAG-BENCH (**Ours**) | **Image** | 🌐 🔺 🌻 🚗 | ✓ | ✓ |

Table 1: Compared with previous works, MRAG-BENCH focuses on evaluating LVLMs in utilizing vision-centric retrieval-augmented multimodal knowledge. "Diverse scenarios" refers to whether a benchmark categorized different scenarios during evaluation. 🌐: Web, 🔺: ImageNet (Russakovsky et al., 2015), 🌻: Flowers102 (Nilsback & Zisserman, 2008), 🚗: StanfordCars (Krause et al., 2013).

# 1 INTRODUCTION

Retrieval-augmented generation (RAG) has emerged as a promising direction in large vision-language models (LVLMs) (OpenAI, 2023; Liu et al., 2023a; Bai et al., 2023; Huang et al., 2023; Chen et al., 2023a; Hu et al., 2024b; Chen et al., 2024; Tong et al., 2024; McKinzie et al., 2024). By incorporating external knowledge during generation, models such as Wiki-LLaVA (Caffagni et al., 2024) have demonstrated improved performance in knowledge-intensive question answering tasks.

There are several existing benchmarks evaluating retrieval-augmented LVLMs. For example, OK-VQA (Marino et al., 2019) focused on scenarios where the image content alone is insufficient to answer the questions. A-OKVQA (Schwenk et al., 2022) further extended this dataset to incorporate additional types of world knowledge. More recent works (Chang et al., 2022; Chen et al., 2023b; Mensink et al., 2023) further expanded and curated large-scale knowledge base data to evaluate pre-trained vision and language models in knowledge-intensive and information-seeking visual questions. However, as shown in Table 1, these benchmarks remain text-centric, as their questions can often be resolved with related external textual knowledge. In contrast, retrieving visual information is sometimes more beneficial than retrieving text, as humans often gain greater insights from it. Specifically, we illustrate examples in Figure 1 where retrieving correct textual knowledge can be *hard* and retrieved textual knowledge can be *useless*, while retrieving additional images is helpful. For instance, when presented with a top-down view of a car, humans may struggle to accurately identify it; however, with a front-facing view, they can quickly recognize the vehicle and effectively leverage the visual information.

In this paper, we introduce MRAG-BENCH, a benchmark specifically designed for vision-centric evaluation for retrieval-augmented multimodal models, with visual questions typically benefit more from retrieving visual knowledge than textual information. MRAG-BENCH consists of 16,130 images and 1,353 human-annotated multi-choice questions spanning 9 distinctive scenarios. Focusing on utilizing visually augmented knowledge in real-world scenarios, we divide our benchmark into two aspects: *perspective*, where changes in visual entity's perspective requiring visually augmented knowledge; and *transformative*, where the visual entity undergoes transformative change physically thus requiring visually augmented knowledge. Specifically, MRAG-BENCH requires models to reason about visual entities that undergo perspective changes, such as *angle*, *partial*, *scope* and *occlusion*, as well as transformative changes, such as *temporal*, *incomplete*, *biological* and *deformations*. Additionally, MRAG-BENCH includes 9,673 human-selected images, which serves as the ground-truth image knowledge corpus for model evaluation.

We conduct extensive experiments on MRAG-BENCH to evaluate 10 open-source and 4 proprietary LVLMs. The results confirm that MRAG-BENCH is vision-centric, as all LVLMs show greater improvements when augmented with images compared to textual knowledge. Our results indicate that the best-performing GPT-4o model only achieve 68.68% and 74.5% of accuracy without RAG knowledge and with ground-truth (GT) RAG knowledge, respectively. This substantially outperforms the best open-source model LLaVA-OneVision by 15.39% and 15.52%, respectively.

| Statistic | Number |
|---|---|
| Total questions | 1,353 |
| - Multiple-choice questions | 1,353 (100%) |
| - Questions newly annotated | 1,353 (100%) |
| Total Scenarios | 9 |
| Unique number of questions | 375 |
| Unique number of answers | 663 |
| Total number of images | 16,130 |
| Unique number of images | 16,130 |
| Human selected images | 9,673 |
| Average image size (px) | 1076 x 851 |
| Maximum question length | 20 |
| Maximum answer length | 9 |
| Average question length | 8.03 |
| Average answer length | 2.16 |
| Average choice number | 4 |

Table 2: Key statistics of MRAG-BENCH.

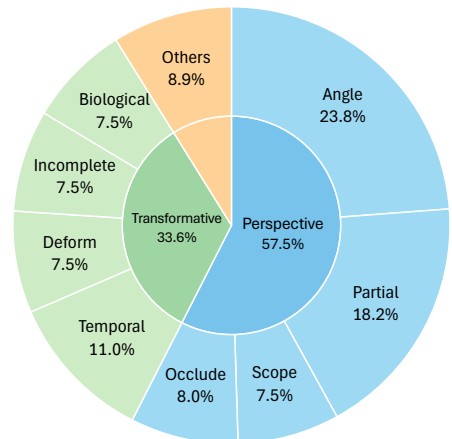

Figure 2: Scenarios distribution of MRAG-BENCH.

Notably, we observe while all models improve with GT knowledge, only proprietary models are able to effectively utilize noisy retrieved multimodal knowledge. This indicates the gap between open-source and close-source models still exists. Open-source models are falling short on their parametric knowledge and the ability to distinguish between high-quality and poor-quality retrieved visually augmented examples. In comparison to humans, GPT-4o achieves only a 5.82% improvement when augmented with GT knowledge and 0.28% with retrieved knowledge, whereas humans demonstrate a 33.16% and 22.91% improvement, respectively. These results highlight the importance of MRAG-BENCH in encouraging the community to develop LVLMs better utilizing of visually augmented knowledge.

## 2 MRAG-BENCH

### 2.1 BENCHMARK OVERVIEW

Our benchmark is designed for systematic evaluation of LVLM's vision-centric multimodal RAG abilities. To achieve this, we focus on evaluating the model's understanding of image objects that are not commonly associated with its knowledge base, while the collected ground-truth images can help incentivize specific visual concepts within LVLMs' memory. Therefore, we divide our benchmark into two main aspects, as illustrated in the examples in Figure 1:

- *perspective*, refers to the challenges in visual recognition and reasoning that arise when a visual entity is presented from varying viewpoints, scopes, or levels of visibility.

- *transformative*, refers to the challenges that arise when a visual entity undergoes fine-grained physical transformations, making it unfamiliar or not easily associated with the model's prior knowledge.

MRAG-BENCH consists of 16,130 images and 1,353 multiple choice questions, with key statistics shown in Table 2. MRAG-BENCH adheres to the following design principles: (1) it focuses on real-world scenarios where visually augmented information is useful; (2) it incorporates 9 diverse multimodal RAG scenarios covering various types of image objects; (3) it features cleaned ground-truth images for each question that align with human knowledge; and (4) it provides robust evaluation settings for deterministic evaluations. Unlike previous works focus on retrieving textual knowledge, evaluation on MRAG-BENCH focuses on retrieving vision-centric knowledge, which can be formulated as follows: Given a query tuple $\mathbf{Q}$ composed of (query image, textual question), the multimodal retriever $\mathcal{R}$ returns a set of relevant images $\mathbf{I}$ ($[\mathbf{i}_1, \mathbf{i}_2, ..., \mathbf{i}_N]$), then the LVLM $\mathcal{M}$ take the input ($\mathbf{Q}, \mathbf{I}$) and output the final answer.

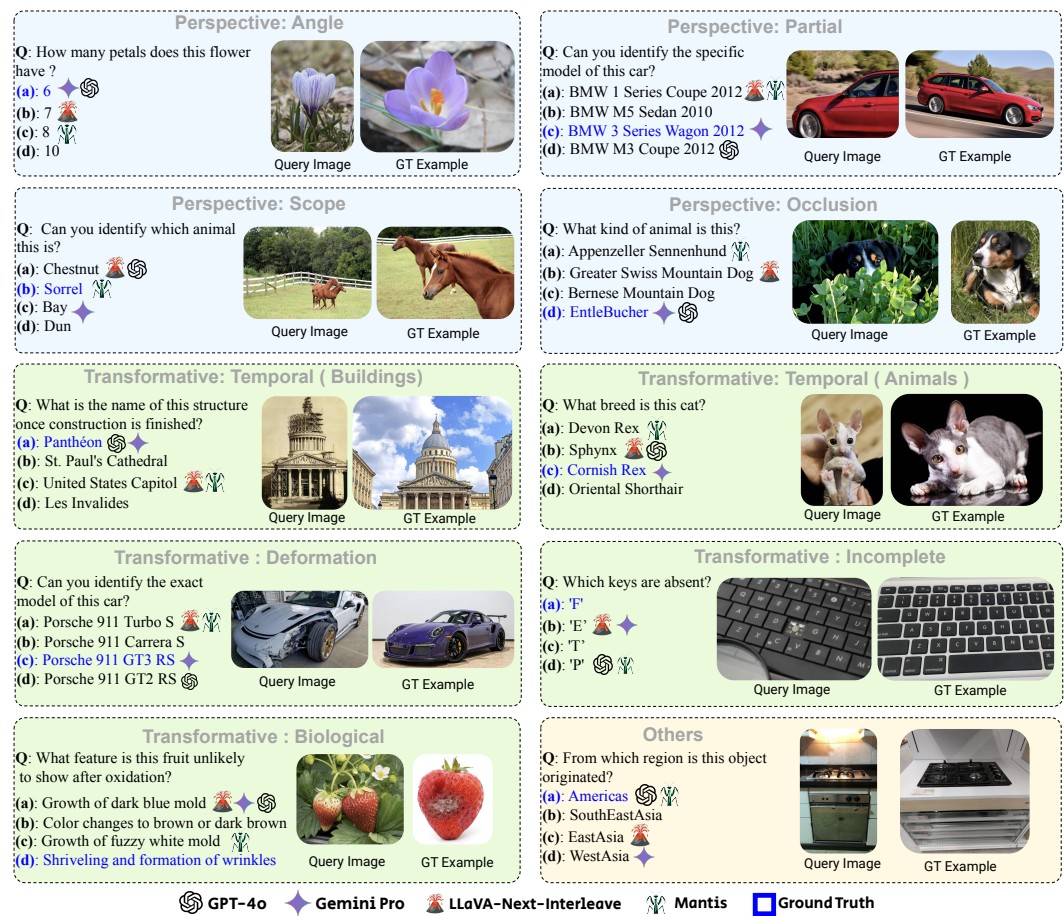

Figure 3: Qualitative examples on MRAG-BENCH. For each scenario, we show the result of GPT-4o (OpenAI, 2023), Gemini Pro (Team et al., 2023), LLaVA-Next-Interleave (Li et al., 2024b) and Mantis-8B-Siglip (Jiang et al., 2024a). The ground-truth answer is in blue.

## 2.2 BENCHMARK COMPOSITION

MRAG-BENCH provides a systematic evaluation across 9 distinctive multimodal RAG scenarios, with four scenarios focused on the *perspective* understanding of visual entities, four on *transformative* understanding, and one categorized as "others". As illustrated in Figure 2, each scenario comprises 7.5% to 23.8% of the whole benchmark. The selected examples of each scenario is shown in Figure 3. The details of each scenario are introduced as follows.

***Perspective* understanding aspect.** First, we have *perspective* aspect comprising [ANGLE], [PARTIAL], [SCOPE], and [OCCLUSION] dimensions.

- [ANGLE] evaluates the ability of models to utilize visual knowledge of common shooting angles to identify and reason about less common, long-tailed viewpoints of visual entities.

- [PARTIAL] evaluates the ability of models to use complete appearance knowledge to identify and reason when only a partial image of the visual entities is available.

- [SCOPE] evaluates the ability of models to leverage high-resolution, detailed images for identifying and reasoning about visual entities in longer-scoped, low-resolution images.

- [OCCLUSION] evaluates the ability of models to use ground-truth image knowledge to identify and reason when visual entities are occluded or partially hidden in natural scenes.

***Transformative* understanding aspect.** On the other hand, the *transformative* understanding scenarios cover [TEMPORAL], [DEFORMATION], [INCOMPLETE], and [BIOLOGICAL] dimensions.

- [TEMPORAL] evaluates the ability of models to use familiar image knowledge to identify and reason about visual entities undergoing temporal changes that may not be represented in the model's knowledge base.
- [DEFORMATION] evaluates the ability of models to use intact physical appearance knowledge to identify and reason when visual entities undergo deformation not captured in the model's knowledge base.
- [INCOMPLETE] evaluates the ability of models to compare and contrast the complete layout and structure of image knowledge to identify and reason about missing parts and the correct layout of visual entities.
- [BIOLOGICAL] evaluates the ability of models to utilize image knowledge after biological transformations of the visual entities.

[OTHERS] aims to evaluate the ability of models to leverage geographic image knowledge to accurately identify and reason about the correct regions of origin for the visual entities of interest. All these scenarios work in tandem to comprehensively evaluate LVLMs' abilities of leveraging visually augmented knowledge.

## 2.3 DATA COLLECTION

As the guidelines discussed in § 2.1, our benchmark collection involves a clean ground-truth image corpus that can resonate with model's internal knowledge and a query question and image that challenge model's memory according to our definition of 9 diverse scenarios. To collect a dataset for systematic evaluation of vision-centric multimodal RAG scenarios, we manually annotate all multiple-choice question answering (MCQA) data while sourcing images from either publicly available datasets or manually scraping them from the web.

**Collection of *perspective* aspect.** To collect diverse image objects and knowledge that are not extensively represented in LVLMs' memories (Zhang et al., 2024c), we considered three sources of data, ImageNet (Russakovsky et al., 2015), Oxford Flowers102 (Nilsback & Zisserman, 2008), and StanfordCars (Krause et al., 2013). To construct a high quality image corpus, for each of the image class that we included in our benchmark, we examined the validation set and excluded the unqualified images which can't provide sufficient visual information for the recognition of this class. Among the selected corpus, we further humanly picked five representative examples covering the diverse aspects of each class object, as the five ground-truth examples in our experimental results (See §3). For constructing the query images, we adhered to our scenario definitions and manually selected qualified images for the [ANGLE], [SCOPE], and [OCCLUSION] scenarios. For the [PARTIAL] scenario, we randomly cropped images by 50% in both height and width. Then we performed another human inspection to ensure the quality of the cropped images, filtering out examples where the visual object did not occupy the dominant area of the image. We repeated the random cropping process until satisfactory images were obtained, filtering to 20.4 GT images per question on average.

**Collection of *transformative* aspect.** We chose to manually scrape images from the web based on the definitions of the *transformative* aspect. To construct the image corpus, we employed Bing Image Search for each of the image object keyword predefined by us, please refer to Appendix A.1 for more details. We filtered out image objects that did not form a clear transformative pair between the query image and the ground-truth image, retaining approximately 74% of the keyword names in the process. For ground-truth image examples, we employed automatic scripts to download the top 15 images related to its keyword names and human filtered out the unqualified image. On average, this results to 5.9 images per question and the five ground-truth images used during our evaluation are manually selected same as in *perspective* aspect.

According to our guidelines, additional related image object knowledge from the same geographic region can assist in identifying that region more effectively. For the [OTHERS] scenario, we source the data from the GeoDE dataset (Ramaswamy et al., 2023). For each distinct image object category,

| Model | Overall | Perspective | | | | Transformative | | | | Others |
|---|---|---|---|---|---|---|---|---|---|---|
| | | Angle | Partial | Scope | Occlusion | Temporal | Deformation | Incomplete | Biological | |
| Random chance | 24.83 | 27.64 | 23.98 | 24.51 | 19.44 | 22.15 | 25.49 | 29.41 | 25.49 | 22.5 |
| Human performance | 38.47 | 25.16 | 34.96 | 31.37 | 41.67 | 21.48 | 24.51 | 58.82 | 54.9 | 53.33 |
| + Retrieved RAG | $61.38_{+22.91}$ | $62.42_{+37.26}$ | $60.16_{+25.2}$ | $58.82_{+27.45}$ | $62.96_{+21.29}$ | $54.36_{+32.88}$ | $49.02_{+24.51}$ | $78.43_{+19.61}$ | $63.73_{+8.83}$ | $62.5_{+9.17}$ |
| + GT RAG | $71.63_{+33.16}$ | $83.85_{+58.69}$ | $70.33_{+35.37}$ | $66.67_{+35.3}$ | $69.44_{+27.77}$ | $59.73_{+38.25}$ | $68.63_{+44.12}$ | $83.33_{+24.51}$ | $73.53_{+18.63}$ | $69.17_{+15.84}$ |
| *Open-Source LVLMs* | | | | | | | | | | |
| OpenFlamingo-v2-9B | 26.83 | 27.95 | 26.02 | 31.37 | 30.56 | 29.53 | 34.31 | 20.59 | 17.65 | 21.67 |
| + Retrieved RAG | $28.31_{+1.48}$ | $29.5_{+1.55}$ | $28.86_{+2.84}$ | $28.43_{-2.94}$ | $30.56_{+0.0}$ | $34.23_{+4.7}$ | $31.37_{-2.94}$ | $22.55_{+1.96}$ | $21.57_{+3.92}$ | $22.5_{+0.83}$ |
| + GT RAG | $28.90_{-2.07}$ | $26.71_{-1.24}$ | $33.74_{+7.72}$ | $28.43_{-2.94}$ | $33.33_{+2.77}$ | $35.57_{+6.04}$ | $27.45_{-6.86}$ | $27.45_{-6.86}$ | $25.49_{+7.84}$ | $18.33_{-3.34}$ |
| Idefics2-8B | 31.04 | 31.06 | 33.33 | 31.37 | 38.89 | 30.2 | 35.29 | 25.49 | 24.51 | 26.67 |
| + Retrieved RAG | $30.16_{-0.88}$ | $29.81_{-1.25}$ | $27.64_{-5.69}$ | $29.41_{-1.96}$ | $36.11_{-2.78}$ | $36.24_{+6.04}$ | $28.43_{-6.86}$ | $27.45_{+1.96}$ | $32.35_{+7.84}$ | $25.83_{-0.84}$ |
| + GT RAG | $37.03_{+5.99}$ | $36.34_{+5.28}$ | $35.37_{+2.04}$ | $38.24_{+6.87}$ | $54.63_{+15.74}$ | $47.65_{+17.45}$ | $36.27_{+0.98}$ | $24.51_{-0.98}$ | $34.31_{+9.8}$ | $25.83_{-0.84}$ |
| VILA1.5-13B | 43.68 | 45.34 | 41.87 | 52.94 | 48.15 | 50.34 | 38.24 | 21.57 | 30.39 | 57.5 |
| + Retrieved RAG | $35.48_{-8.2}$ | $33.54_{-11.8}$ | $28.86_{-13.01}$ | $29.41_{-23.53}$ | $40.74_{-7.41}$ | $47.65_{-2.69}$ | $33.33_{-4.91}$ | $22.55_{+0.98}$ | $33.33_{+2.94}$ | $54.17_{-3.33}$ |
| + GT RAG | $47.01_{+3.33}$ | $45.65_{+0.31}$ | $46.75_{+4.88}$ | $39.22_{-13.72}$ | $51.85_{+3.7}$ | $53.69_{+3.35}$ | $43.14_{+4.9}$ | $25.49_{+3.92}$ | $44.12_{+13.73}$ | $69.17_{+11.67}$ |
| Mantis-8B-clip-llama3 | 40.8 | 45.03 | 39.43 | 42.16 | 49.07 | 49.66 | 36.27 | 28.43 | 19.61 | 45.0 |
| + Retrieved RAG | $36.88_{-3.92}$ | $36.65_{-8.38}$ | $34.96_{-4.47}$ | $42.16_{0.0}$ | $47.22_{-1.85}$ | $50.34_{+0.68}$ | $33.33_{-2.94}$ | $18.63_{-9.8}$ | $21.57_{+1.96}$ | $42.5_{-2.5}$ |
| + GT RAG | $44.72_{+3.92}$ | $48.14_{+3.11}$ | $46.75_{+7.32}$ | $43.14_{+0.98}$ | $54.63_{+5.56}$ | $57.05_{+7.39}$ | $45.1_{+8.83}$ | $19.61_{-8.82}$ | $18.63_{-0.98}$ | $51.67_{+6.67}$ |
| Mantis-8B-siglip-llama3 | 45.01 | 46.89 | 45.12 | 57.84 | 58.33 | 45.64 | 45.1 | 26.47 | 29.41 | 45.0 |
| + Retrieved RAG | $39.62_{-5.39}$ | $42.55_{-4.34}$ | $35.37_{-9.75}$ | $47.06_{-10.78}$ | $47.22_{-11.11}$ | $42.95_{-2.69}$ | $45.1_{0.0}$ | $23.53_{-2.94}$ | $29.41_{0.0}$ | $40.83_{-4.17}$ |
| + GT RAG | $48.85_{+3.84}$ | $54.66_{+7.77}$ | $52.85_{+7.73}$ | $51.96_{-5.88}$ | $58.33_{0.0}$ | $48.99_{+3.35}$ | $50.0_{+4.9}$ | $21.57_{-4.9}$ | $33.33_{+3.92}$ | $49.17_{+4.17}$ |
| Deepseek-VL-7B-chat | 43.39 | 45.34 | 47.56 | 47.06 | 45.37 | 46.31 | 48.04 | 28.43 | 20.59 | 49.17 |
| + Retrieved RAG | $34.66_{-8.73}$ | $33.54_{-11.8}$ | $32.11_{-15.45}$ | $33.33_{-13.73}$ | $37.04_{-8.33}$ | $43.62_{-2.69}$ | $40.2_{-7.84}$ | $20.59_{-7.84}$ | $26.47_{+5.88}$ | $45.0_{-4.17}$ |
| + GT RAG | $50.33_{+6.94}$ | $54.04_{+8.7}$ | $56.5_{+8.94}$ | $50.98_{+3.92}$ | $56.48_{+11.11}$ | $57.05_{+10.74}$ | $50.0_{+1.96}$ | $21.57_{-6.86}$ | $23.53_{+2.94}$ | $60.83_{+11.66}$ |
| LLaVA-NeXT-Interleave-7B | 43.46 | 44.41 | 43.5 | 40.2 | 64.81 | 44.97 | 44.12 | 32.35 | 26.47 | 45.83 |
| + Retrieved RAG | $40.35_{-3.11}$ | $40.06_{-4.35}$ | $33.33_{-10.17}$ | $39.22_{-0.98}$ | $56.48_{-8.33}$ | $43.62_{-1.35}$ | $44.12_{+0.0}$ | $27.45_{-4.9}$ | $36.27_{+9.8}$ | $49.17_{+3.34}$ |
| + GT RAG | $52.99_{+9.53}$ | $54.97_{+10.56}$ | $54.88_{+11.38}$ | $49.02_{+8.82}$ | $62.04_{-2.77}$ | $52.35_{+7.38}$ | $47.06_{+2.94}$ | $38.24_{+5.89}$ | $48.04_{+21.57}$ | $61.67_{+15.84}$ |
| mPLUG-Owl3-7B | 49.74 | 48.45 | 50.81 | 54.9 | 58.33 | 54.36 | 51.96 | 30.39 | 45.1 | 51.67 |
| + Retrieved RAG | $41.83_{-7.91}$ | $40.06_{-8.39}$ | $36.59_{-14.22}$ | $40.2_{-14.7}$ | $50.0_{-8.33}$ | $50.34_{-4.02}$ | $46.08_{-5.88}$ | $20.59_{-9.8}$ | $51.96_{+6.86}$ | $46.67_{-5.0}$ |
| + GT RAG | $56.32_{+6.58}$ | $58.39_{+9.94}$ | $58.94_{+8.13}$ | $58.82_{+3.92}$ | $62.96_{+4.63}$ | $61.74_{+7.38}$ | $59.8_{+7.84}$ | $26.47_{-3.92}$ | $50.0_{+4.9}$ | $58.33_{+6.66}$ |
| LLaVA-OneVision | 53.29 | 58.39 | 56.1 | 49.02 | 60.19 | 47.65 | 53.92 | 37.25 | 52.94 | 51.67 |
| + Retrieved RAG | $50.11_{-3.18}$ | $50.93_{-7.46}$ | $48.78_{-7.32}$ | $50.0_{+0.98}$ | $60.19_{+0.0}$ | $50.34_{+2.69}$ | $48.04_{-5.88}$ | $33.33_{-3.92}$ | $53.92_{+0.98}$ | $54.17_{+2.5}$ |
| + GT RAG | $58.98_{+5.69}$ | $62.42_{+4.03}$ | $63.82_{+7.72}$ | $59.8_{+10.78}$ | $66.67_{+6.48}$ | $59.73_{+12.08}$ | $53.92_{+0.0}$ | $30.39_{-6.86}$ | $57.84_{+4.9}$ | $60.83_{+9.16}$ |
| Pixtral-12B | 47.97 | 52.48 | 45.53 | 58.82 | 50.0 | 51.68 | 49.02 | 38.24 | 42.16 | 37.5 |
| + Retrieved RAG | $45.97_{-2.0}$ | $51.86_{-0.62}$ | $40.24_{-5.29}$ | $53.92_{-4.9}$ | $50.93_{+0.93}$ | $49.66_{-2.02}$ | $47.06_{-1.96}$ | $19.61_{-18.63}$ | $47.06_{+4.9}$ | $46.67_{+9.17}$ |
| + GT RAG | $59.28_{+11.31}$ | $63.04_{+10.56}$ | $63.41_{+17.88}$ | $65.69_{+6.87}$ | $66.67_{+16.67}$ | $61.74_{+10.06}$ | $59.8_{+10.78}$ | $20.59_{-17.65}$ | $50.98_{+8.82}$ | $65.0_{+27.5}$ |
| *Proprietary LVLMs* | | | | | | | | | | |
| GPT-4-Turbo | 57.21 | 64.29 | 59.35 | 54.9 | 56.48 | 62.42 | 47.06 | 41.18 | 59.8 | 50.0 |
| + Retrieved RAG | $58.95_{+1.74}$ | $66.53_{+2.24}$ | $59.94_{+0.59}$ | $53.94_{-0.96}$ | $66.74_{+10.26}$ | $59.73_{-2.69}$ | $49.06_{+2.0}$ | $38.27_{-2.91}$ | $62.78_{+2.98}$ | $58.83_{+8.83}$ |
| + GT RAG | $62.85_{+5.64}$ | $68.94_{+4.65}$ | $69.51_{+10.16}$ | $60.78_{+5.88}$ | $67.59_{+11.11}$ | $63.33_{+0.91}$ | $51.96_{+4.9}$ | $38.24_{-2.94}$ | $59.8_{+0.0}$ | $62.5_{+12.5}$ |
| Gemini Pro | 61.71 | 68.01 | 69.92 | 73.53 | 71.3 | 70.47 | 42.16 | 39.22 | 53.92 | 40.83 |
| + Retrieved RAG | $65.93_{+4.22}$ | $73.29_{+5.28}$ | $69.92_{+0.0}$ | $69.61_{-3.92}$ | $73.15_{+1.85}$ | $75.84_{+5.37}$ | $49.02_{+6.86}$ | $34.31_{-4.91}$ | $56.86_{+2.94}$ | $65.0_{+24.17}$ |
| + GT RAG | $71.40_{+9.69}$ | $77.33_{+9.32}$ | $79.27_{+9.35}$ | $78.43_{+4.9}$ | $75.93_{+4.63}$ | $78.52_{+8.05}$ | $54.9_{+12.74}$ | $36.27_{-2.95}$ | $61.76_{+7.84}$ | $72.5_{+31.67}$ |
| Claude 3.5 Sonnet | 59.87 | 70.19 | 57.72 | 56.86 | 57.41 | 68.46 | 48.04 | 49.02 | 62.75 | 47.5 |
| + Retrieved RAG | $63.56_{+3.69}$ | $73.91_{+3.72}$ | $70.73_{+13.01}$ | $56.86_{+0.0}$ | $62.96_{+5.55}$ | $70.47_{+2.01}$ | $55.88_{+7.84}$ | $31.37_{-17.65}$ | $62.75_{+0.0}$ | $53.33_{+5.83}$ |
| + GT RAG | $71.10_{+11.23}$ | $78.88_{+8.69}$ | $80.49_{+22.77}$ | $76.47_{+19.61}$ | $70.37_{+12.96}$ | $75.17_{+6.71}$ | $67.65_{+19.61}$ | $36.27_{-12.75}$ | $65.69_{+2.94}$ | $59.17_{+11.67}$ |
| GPT-4o | 68.68 | 76.09 | 70.42 | 69.61 | 74.07 | 73.82 | 61.21 | 47.62 | 58.82 | 65.83 |
| + Retrieved RAG | $68.96_{+0.28}$ | $77.95_{+1.86}$ | $78.86_{+8.44}$ | $69.61_{+0.0}$ | $75.0_{+0.93}$ | $73.83_{+0.01}$ | $54.9_{+7.28}$ | $26.47_{-34.74}$ | $59.8_{+0.98}$ | $68.33_{+2.5}$ |
| + GT RAG | $74.50_{+5.82}$ | $84.47_{+8.38}$ | $77.46_{+7.04}$ | $82.35_{+12.74}$ | $79.63_{+5.56}$ | $77.18_{+3.36}$ | $68.62_{+7.41}$ | $30.95_{-16.67}$ | $62.75_{+3.93}$ | $80.0_{+14.17}$ |

Table 3: Accuracy scores on MRAG-BENCH. The highest scores for open-source models in each section and proprietary models are highlighted in blue and red, respectively. CLIP retriever is consistently used across all models. Both Retrieved RAG and GT RAG employ top-5 image examples (except for the incomplete scenario, where a single example is intuitively sufficient). The relative difference in performance compared to the score without RAG is shown in subscript, with blue indicating performance drops and red indicating improvements.

we randomly sampled 3 out of 6 regions to serve as the answers for each question and selected the corresponding image as the query image.

**Quality control.** After constructing the entire benchmark, we implemented two quality control procedures: an automatic check with predefined rules and a manual examination of each instance. The automatic check verifies the correct MCQA format, assesses image validity and filters out redundant images in the corpus, more details are presented in Appendix A.1. The manual examination is conducted by two experts in this field, who checked the correspondence between query images and ground-truth image examples, and filtered or revised ambiguous questions and uncorrelated query image and ground-truth images.

## 3  EXPERIMENTS

In this section, we first introduce the experimental setup and evaluation metric (§ 3.1). Then, we present a comprehensive evaluation of 14 recent LVLMs (§ 3.2). We demonstrate the importance of visual knowledge and discuss the critical findings revealed by the results from MRAG-BENCH.

### 3.1  EXPERIMENTAL SETUP

We evaluate 14 popular LVLMs on MRAG-BENCH, including 4 proprietary models and 10 open-sourced models that can accept multi-image inputs:

- **Proprietary models**: GPT-4o (0513) (OpenAI, 2023), GPT-4-Turbo (OpenAI, 2023), Gemini Pro (Team et al., 2023), and Claude 3.5 Sonnet (Anthropic, 2024).
- **Open-source models**: OpenFlamingo (v2-9B) (Awadalla et al., 2023), Idefics (v2-8B) (Laurençon et al., 2024), VILA (v1.5-13B) (Lin et al., 2023), LLaVA-NeXT-Interleave-7B (Li et al., 2024b), LLaVA-OneVision (Li et al., 2024a), Mantis (clip-llama3, and siglip-llama3 versions; 8B) (Jiang et al., 2024a), mPLUG-Owl3-7B (Ye et al., 2024), Deepseek-VL-7B-chat (Lu et al., 2024a), and Pixtral-12B (Team, 2024).

**Evaluation setup.**  We follow standard MCQA evaluation setup and employ accuracy score as our metric. We adopt default generation hyper-parameters selected by each model. Following Lu et al. (2024b), we employ GPT-3.5-turbo to extract the multiple choice answer in rare cases where our pre-defined automatic extraction rules failed. We refer the readers to Appendix A.1 and B for more details on evaluation prompts for both without multimodal RAG and with multimodal RAG scenarios, answer extraction prompt and human performance evaluation protocol.

### 3.2  MAIN RESULTS

As shown in Table 3, the average performance of the most advanced LVLMs is not better than 68.68% without multimodal RAG knowlege, and 74.5% with ground-truth knowledge, which demonstrates MRAG-BENCH to be a challenging benchmark. The mean accuracies of open-source LVLMs are between 26.83% and 53.29% without RAG knowledge and between 28.90% and 59.28% with ground-truth knowledge, which fall behind from advanced proprietary LVLMs. Notably, MRAG-BENCH proves to be knowledge-intensive as average humans achieved 38.47% without RAG knowledge, while proprietary LVLMs generally perform well, suggesting that their extensive training data equips them with a broader knowledge base. However, when provided with either retrieved or ground-truth knowledge, humans achieve the most significant improvements of 22.91% and 33.16%, respectively. This underscore the need of LVLMs to better utilize visually augmented information like humans.

**Can LVLMs utilize retrieved and ground-truth image knowledge well?**  As illustrated in Table 3, all models demonstrate improvement when ground-truth image RAG knowledge is provided. Among the open-source models, they achieve improvements ranging from 2.07% to 11.31% when using ground-truth RAG knowledge, whereas 5.64% to 9.69% improvements are observed from proprietary LVLMs. Interestingly, when images from the multimodal retriever is provided, almost all open-source LVLMs on average demonstrate a declined performance while proprietary models can still gain improvement. This indicate proprietary models possess emerging abilities to distinguish between good and bad image knowledge sources, which is a critical skill in the multimodal RAG domain. We further conducted a qualitative analysis to investigate the reasons behind this, as detailed in the following paragraphs.

**Fine-grained results.**  We also report fine-grained scores across 9 scenarios on MRAG-BENCH in Table 3. Remarkably, GPT-4o surpasses most other baselines in various categories, with exceptions in problems related to partial, incomplete and biological scenarios. Notably, GPT-4o outperforms human performance on all perspective aspect as well as on temporal and deformation scenarios within the transformative aspect. We conjecture that incomplete and biological scenarios are less likely to be included in the training knowledge. Interestingly, all models exhibit a decline in performance on incomplete scenarios, with only a few exceptions, while humans find this

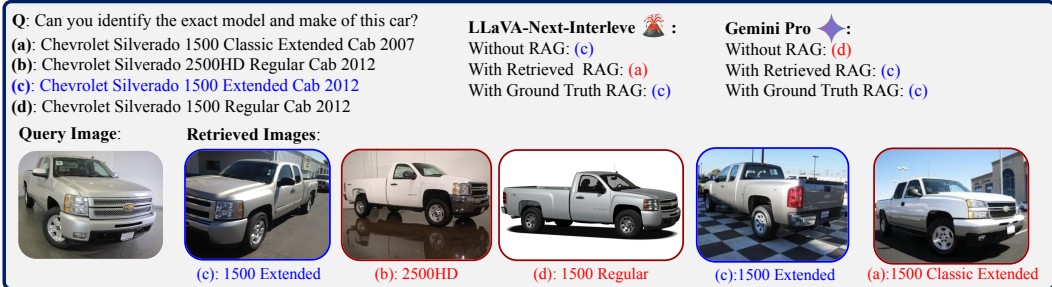

Figure 4: Qualitative Example of Proprietary model (Gemini Pro) identifies and utilizes correct examples, while open-source model (LLaVA-Next-Interleave) is misled by noisy retrieved information, resulting in incorrect answers.

| Model | Overall | Perspective | | | | Transformative | | | | Others |
|---|---|---|---|---|---|---|---|---|---|---|
| | | Angle | Partial | Scope | Occlusion | Temporal | Deformation | Incomplete | Biological | |
| LLaVA-NeXT-Interleave-7B | 43.46 | 44.41 | 43.5 | 40.2 | 64.81 | 44.97 | 44.12 | 32.35 | 26.47 | 45.83 |
| + Retrieved Text RAG | $37.99_{-5.47}$ | $37.58_{-6.83}$ | $34.96_{-8.54}$ | $33.33_{-6.87}$ | $50.0_{-14.81}$ | $41.61_{-3.36}$ | $35.29_{-8.83}$ | $30.39_{-1.96}$ | $27.45_{+0.98}$ | $51.67_{+5.84}$ |
| + Retrieved Image RAG | $40.35_{-3.11}$ | $40.06_{-4.35}$ | $33.33_{-10.17}$ | $39.22_{-0.98}$ | $56.48_{-8.33}$ | $43.62_{-1.35}$ | $44.12_{+0.0}$ | $27.45_{-4.9}$ | $36.27_{+9.8}$ | $49.17_{+3.34}$ |
| + GT Text RAG | $41.09_{-2.37}$ | $41.93_{-2.48}$ | $39.02_{-4.48}$ | $38.24_{-1.96}$ | $56.48_{-8.33}$ | $44.97_{+0.0}$ | $43.14_{-0.98}$ | $30.39_{-1.96}$ | $21.57_{-4.9}$ | $50.83_{+5.0}$ |
| + GT Image RAG | $52.99_{+9.53}$ | $54.97_{+10.56}$ | $54.88_{+11.38}$ | $49.02_{+8.82}$ | $62.04_{-2.77}$ | $52.35_{+7.38}$ | $47.06_{+2.94}$ | $38.24_{+5.89}$ | $48.04_{+21.57}$ | $61.67_{+15.84}$ |
| + GT Image & Text RAG | $47.82_{+4.36}$ | $47.83_{+3.42}$ | $48.78_{+5.28}$ | $44.12_{+3.92}$ | $58.33_{-6.48}$ | $49.66_{+4.69}$ | $48.04_{+3.92}$ | $30.39_{-1.96}$ | $35.29_{+8.82}$ | $62.5_{+16.67}$ |
| GPT-4-Turbo | 57.21 | 64.29 | 59.35 | 54.9 | 56.48 | 62.42 | 47.06 | 41.18 | 59.8 | 50.0 |
| + Retrieved Text RAG | $56.61_{-0.6}$ | $61.8_{-2.49}$ | $59.35_{+0.0}$ | $59.8_{+4.9}$ | $58.33_{+1.85}$ | $59.06_{-3.36}$ | $49.02_{+1.96}$ | $33.33_{-7.85}$ | $60.78_{+0.98}$ | $52.5_{+2.5}$ |
| + Retrieved Image RAG | $58.95_{+1.74}$ | $66.53_{+2.24}$ | $59.94_{+0.59}$ | $53.94_{-0.96}$ | $66.74_{+10.26}$ | $59.73_{-2.69}$ | $49.06_{+2.0}$ | $38.27_{-2.91}$ | $62.78_{+2.98}$ | $58.83_{+8.83}$ |
| + GT Text RAG | $58.98_{+1.77}$ | $68.01_{+3.72}$ | $63.41_{+4.06}$ | $65.69_{+10.79}$ | $63.89_{+7.41}$ | $59.73_{-2.69}$ | $38.24_{-8.82}$ | $37.25_{-3.93}$ | $58.82_{-0.98}$ | $50.83_{+0.83}$ |
| + GT Image RAG | $62.85_{+5.64}$ | $68.94_{+4.65}$ | $69.51_{+10.16}$ | $60.78_{+5.88}$ | $67.59_{+11.11}$ | $63.33_{+0.91}$ | $51.96_{+4.9}$ | $38.24_{-2.94}$ | $59.8_{+0.0}$ | $62.5_{+12.5}$ |
| + GT Image & Text RAG | $65.11_{+7.9}$ | $72.05_{+7.76}$ | $72.76_{+13.41}$ | $67.65_{+12.75}$ | $70.37_{+13.89}$ | $71.81_{+9.39}$ | $46.08_{-0.98}$ | $39.22_{-1.96}$ | $60.78_{+0.98}$ | $57.5_{+7.5}$ |

Table 4: LVLMs performance on MRAG-BENCH with textual knowledge v.s visual knowledge. Both the open-source and proprietary model benefit more from image knowledge.

task relatively easy, achieving 58.82% and 83.33% scores with ground-truth knowledge. This further highlights the importance of leveraging retrieved visually augmented knowledge to address questions that do not directly incentivize knowledge stored in the models' memories.

**Why can proprietary models better utilize retrieved images?** We conduct an error analysis on an open-source model (LLaVA-Next-Interleave) and a proprietary model (Gemini Pro). For a fair comparison, we filtered results where LLaVA-Next-Interleave answered correctly without or with GT knowledge but was misled to wrong answer with retrieved examples. One example is illustrated in Figure 4, the retrieved images contain two correct examples and three false examples. While Gemini Pro is able to utilize all retrieved images, LLaVA-Next-Interleave leverages bad examples and makes wrong prediction. This example helps explain why do almost all open-source models have lower performance with retrieved knowledge.

## 4 ANALYSIS

In this section, we conduct quantitative analysis addressing three important questions: 1) To what extent can LVLMs benefit more from visual knowledge than from textual knowledge on MRAG-BENCH? (§ 4.1) 2) How does the performance of LVLMs vary with examples retrieved from different retrievers? (§ 4.2) 3) How many ground-truth visual knowledge examples are required for LVLMs to continue benefiting? (§ 4.3)

### 4.1 HOW MUCH CAN VISUAL KNOWLEDGE BENEFIT MORE THAN TEXTUAL KNOWLEDGE?

We used the Wikipedia corpus as of 2023/07/01 as our text knowledge corpus[1]. To ensure a fair comparison, we employed the same multimodal retriever (CLIP) for retrieving either text or image knowledge. The top-5 ranked documents or images are used for augmenting the input. We selected one open-source (LLaVA-Next-Interleave) and one proprietary (GPT-4-Turbo) LVLM to examine their preference for textual knowledge versus image knowledge on MRAG-BENCH. As shown

---

[1]https://www.kaggle.com/datasets/jjinho/wikipedia-20230701

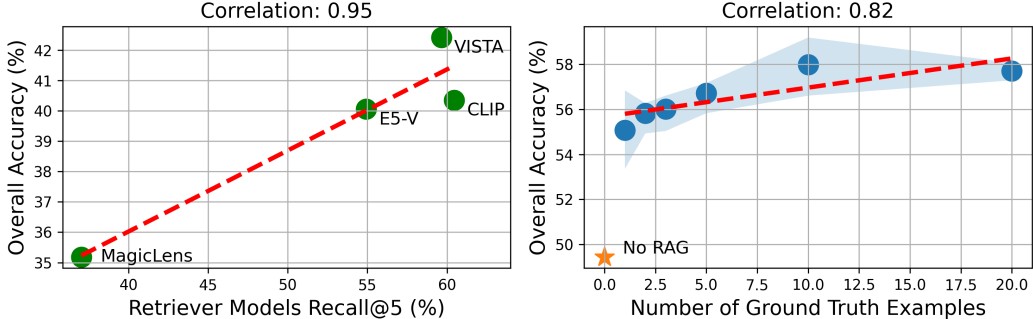

Figure 5: Left: LLaVA-Next-Interleave results with 4 different multimodal retrievers. Its performance using retrieved images correlates 95% with retriever's Recall@5 scores. Right: Average results of three random seed runs. Improve the number of ground-truth RAG examples shows steady increase of model's performance, reaches the maximum with 10 examples.

in Table 4, when both models utilized retrieved knowledge, LLaVA-Next-Interleave demonstrated a 2.36% improvement with image knowledge over text knowledge, while GPT-4-Turbo showed a 2.34% improvement. When using GT knowledge, LLaVA-Next-Interleave exhibited an 11.90% improvement with image knowledge over text knowledge, compared to a 3.87% improvement for GPT-4-Turbo. Interestingly, when both GT image and text knowledge are provided, LLaVA-Next-Interleave indicated less improvement than with GT image alone whereas GPT-4-Turbo further pushed its performance. All these results demonstrate that retrieving visual knowledge is more helpful than retrieving text on MRAG-BENCH.

## 4.2 HOW DOES RETRIEVER PERFORMANCE AFFECT LVLMS?

We picked four recent best-performing multimodal retrievers, including CLIP (Radford et al., 2021), MagicLens (Zhang et al., 2024a), E5-V (Jiang et al., 2024b), VISTA (Zhou et al., 2024) and evaluated their performance (Recall@5). The detailed retriever performance can be found at Table 6 in Appendix C. We selected LLaVA-Next-Interleave as the end model to assess its performance. As shown in Figure 5, when retrievers achieve higher Recall@5 scores (i.e., better retrieved examples), the LVLM's accuracy tends to improve, demonstrating a strong 95% positive correlation. Interestingly, despite similar Recall@5 scores from CLIP and VISTA retrievers, LLaVA-Next-Interleave demonstrated a 2.07% gap in overall accuracy. We conjecture that the order of the correctly retrieved examples may also impact the model's final performance. The sensitivity to the order of retrieved examples is a common issue that persists across various models. Although this phenomenon, known as position bias, has been examined in text-based RAG (Lu et al., 2022b; Wang et al., 2023), its impact on visual RAG remains unexplored, presenting a promising direction for future research.

## 4.3 HOW MANY GROUND-TRUTH IMAGE EXAMPLES ARE NEEDED?

For simplicity, all our experiments used five retrieved or ground-truth image examples. However, it is worth exploring how many examples LVLMs can effectively leverage. As noted in § 2.3, the perspective aspect of our benchmark includes an average of 20.4 ground-truth examples. To investigate further, we perform an analysis focusing on the perspective and others aspects, covering a total of 892 questions. As shown in Figure 5, we evaluated LLaVA-Next-Interleave using 1, 2, 3, 5, 10, 20 GT examples, averaging the results across three random seeds for sampling the GT examples. LLaVA-Next-Interleave saw the greatest improvement of 5.64% with just one GT example. Performance continued to increase steadily, reaching a peak at 10 GT examples, which was 0.29% higher than with 20 GT examples. One possible explanation could be LLaVA-Next-Interleave may not able to better leverage visually augmented knowledge in long context scenarios. Moreover, the complexity of questions affects the number of images needed too, one ground-truth example sometimes help the model the most on MRAG-BENCH. We encourage the research on adaptively deciding the number of necessary images based on the complexity of questions.

## 5 RELATED WORK

We overview three lines of related work: 1) multimodal retrieval-augmented generation benchmarks (§ 5.1), 2) large vision language models (§ 5.2), and 3) retrieval-augmented mutlimodal large language models (§ 5.3).

### 5.1 MULTIMODAL RETRIEVAL-AUGMENTED GENERATION BENCHMARKS

A number of recent benchmarks have been developed to comprehensively assess the capabilities of LVLMs (Lu et al., 2021; 2022a; Li et al., 2023; Liu et al., 2023c; Lu et al., 2024b; Yue et al., 2023; Zhang et al., 2024b; Ying et al., 2024). There are several benchmarks well-suited for evaluating retrieval-augmented LVLMs. For instance, OK-VQA (Marino et al., 2019) and A-OKVQA (Schwenk et al., 2022) both focus on scenarios where external textual knowledge is required to answer visual questions. More recent works (Chang et al., 2022; Chen et al., 2023b; Mensink et al., 2023) have curated large-scale knowledge bases to evaluate models on knowledge-intensive and information-seeking visual questions. In contrast, MRAG-BENCH focus on scenarios where retrieving visual information is more helpful than retrieving text.

### 5.2 LARGE VISION LANGUAGE MODELS

Large Vision Language Models (LVLMs) (Liu et al., 2023b; Zhu et al., 2023a; Dai et al., 2023; Yin et al., 2023; Hu et al., 2024a) have showcased promising results on a wide variety of vision-language tasks. Many works, such as Flamingo (Alayrac et al., 2022), Emu (Sun et al., 2023), Idefics (Laurençon et al., 2023), and VILA (Lin et al., 2023), have demonstrated in-context learning capabilities, where multiple image examples can be leveraged to improve text generation. Recent works start training LVLMs with interleaved image-text corpora, such as MMC4 (Zhu et al., 2023b) and OBELICS (Laurençon et al., 2023), for pretraining, as well as high-quality instruction tuning in models like Mantis-Instruct (Jiang et al., 2024a), LLaVA-Next-Interleave (Li et al., 2024b), and LLaVA-OneVision (Li et al., 2024a), enabling models to process and understand information from multiple images. Naturally, evaluating the ability of LVLMs to effectively leverage visually augmented knowledge becomes an important task, which is the primary focus of MRAG-BENCH.

### 5.3 RETRIEVAL-AUGMENTED MULTIMODAL LARGE LANGUAGE MODELS

Retrieval-augmented generation (RAG) has emerged as a potential solution to overcome limitations in language models by incorporating external knowledge retrieval during the generation process (Lewis et al., 2020; Shi et al., 2024). Reasonably, several works have focused on using multimodal knowledge to enhance the generation capabilities of Large Language Models (LLMs) (Yasunaga et al., 2023; Chen et al., 2022; Zhao et al., 2023; Cui et al., 2024). Recently, more works (Caffagni et al., 2024; Xuan et al., 2024; Du et al., 2024) has incorporated external knowledge to improve LVLMs' general generation abilities and the comprehensiveness of their reasoning. Although some works (Chen et al., 2022; Yuan et al., 2023) have proposed directly using image information from the web, a systematic vision-centric benchmark to evaluate LVLMs' abilities to leverage visually augmented knowledge is lacking, which is the focus of our work.

## 6 CONCLUSION

In this work, we introduce MRAG-BENCH, a benchmark specifically designed for vision-centric evaluation for retrieval-augmented multimodal models. Our evaluation of 14 LVLMs highlights that visually augmented knowledge brings more improvements on MRAG-BENCH compared to textual knowledge. Moreover, the top-performing model, GPT-4o, struggles to effectively utilize the retrieved knowledge, achieving only a 5.82% improvement when augmented with relevant information, compared to a 33.16% improvement demonstrated by human participants. We further conduct extensive analysis and propose several promising directions for future research. Our findings underscore the significance of MRAG-BENCH in motivating the community to develop LVLMs that better utilize retrieved visual knowledge.

ACKNOWLEDGMENTS

We would like to thank the reviewers for their valuable reviews. We would also like to thank UCLA-NLP members for their helpful comments and thank PLUSLab members for their proofreading during the paper clinic session. This research is supported in part by the ECOLE program under Cooperative Agreement HR00112390060 with the US Defense Advanced Research Projects Agency (DARPA), and UCLA-Amazon Science Hub for Humanity and Artificial Intelligence.

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

CONTENTS

## A   MRAG-BENCH DETAILS

### A.1   DATASET CURATION DETAILS

**Dataset collection of transformative aspect**    We chose to manually scrape images from the web based on the definitions of the transformative aspect. To construct the image corpus, we employed Bing Image Search for each of the image object keyword predefined by us. We filtered some of the search results where the image objects do not have a clear pair of query image and ground-truth image example, around 74% keyword names were kept during this process. Here we listed all the keywords that are already filtered and used for search of query image except in biological scenario, it's for search of ground-truth image example. Each search keyword is composed of an "image

object" and a "condition". For example, "A young kitten image of Himalayan Cat", here Himalayan Cat is the image object and a young kitten is the condition. For each of keyword listed below, we searched again for its ground-truth examples (except for biological scenario, it's for query images), in which only "image object" is kept and "conditon" is removed. All searched results are further picked and downloaded by humans to ensure quality. Here is a list of the filtered keywords for transformative aspect:

**Transformative: Temporal**
- A young kitten image of Himalayan Cat
- A young kitten image of Chartreux
- A young kitten image of Burmese
- A young kitten image of Turkish Van
- A young kitten image of American Shorthair
- A young kitten image of British Shorthair
- A young kitten image of Maine Coon
- A young kitten image of Burma (Myanmar)
- A young kitten image of Selkirk Rex
- A young kitten image of Siberian
- A young kitten image of Persian
- A young kitten image of Manx
- A young kitten image of Ocicat
- A young kitten image of Russian Blue
- A young kitten image of Bengal Cat
- A young kitten image of Devon Rex
- A young kitten image of American Bobtail
- A young kitten image of Balinese
- A young kitten image of LaPerm
- A young kitten image of Egyptian Mau
- A young kitten image of Japanese Bobtail
- A young kitten image of Ragdoll
- A young kitten image of Abyssinian
- A young kitten image of American Wirehair
- A young kitten image of Oriental Shorthair
- A young kitten image of Cornish Rex
- A young kitten image of Kurilian Bobtail
- A young kitten image of Singapura Cat
- A young kitten image of Birman
- A young kitten image of Burmilla
- A young kitten image of Korat
- A young kitten image of Tonkinese
- A young kitten image of Somali Cat
- A young kitten image of Norwegian Forest Cat
- A young kitten image of Turkish Angora
- A young kitten image of Siamese
- A picture of Sainte-Chapelle under construction
- A picture of Washington Monument under construction
- A picture of Hearst Castle under construction
- A picture of Time Square under construction
- A picture of Wrigley Building under construction
- A picture of Eiffel Tower under construction
- A picture of The Arc de Triomphe under construction
- A picture of Golden Gate Bridge under construction
- A picture of White House under construction
- A picture of Palace of Versailles under construction
- A picture of Opéra Garnier under construction
- A picture of San Simeon under construction
- A picture of The Louvre under construction
- A picture of Cathédrale Notre-Dame de Paris under construction
- A picture of Sacré-Cœur Basilica under construction

- A picture of Brooklyn Bridge under construction
- A picture of Panthéon under construction
- A picture of Capitol Building under construction
- A picture of Independence Hall under construction
- A picture of Mont Saint-Michel under construction
- A picture of St Patrick's Cathedral under construction
- A picture of Space Needle under construction
- A picture of Château de Chambord under construction
- A picture of Versailles under construction

**Transformative: Deformation**
- An image of Toyota Camry damaged
- An image of Ford F-150 damaged
- An image of Ferrari 458 damaged
- An image of Audi Q5 damaged
- An image of Lamborghini LP640 damaged
- An image of McLaren 675LT damaged
- An image of Mercedes SLC damaged
- An image of Lamborghini Aventador damaged
- An image of Lamborghini LP570 damaged
- An image of Porsche 911 GT3 RS damaged
- An image of Audi A6 damaged
- An image of Audi A4 damaged
- An image of Lamborghini Aventador SV damaged
- An image of GMC Sierra 2500 HD damaged
- An image of Infiniti G37 damaged
- An image of GMC Yukon damaged
- An image of Honda Accord damaged
- An image of Infiniti FX35 damaged
- An image of Tesla Model 3 damaged
- An image of Acura RDX 2020 damaged
- An image of BMW 7 Series damaged
- An image of Audi A5 Sportback damaged
- An image of Hyundai IX35 damaged
- An image of Cadillac XTS damaged
- An image of BMW M3 damaged
- An image of Acura MDX damaged
- An image of Audi A3 damaged
- An image of BMW X3 damaged
- An image of Porsche Boxster damaged
- An image of Mercedes CLA45 AMG damaged
- An image of Jaguar XJ damaged

**Transformative: Incomplete**
- MacBook Keyboard missing keys
- Windows Keyboard missing keys
- Laptop Keyboards (Generic) missing keys
- Mechanical Keyboard missing keys
- Ergonomic Keyboard missing keys
- Compact Keyboard missing keys
- Gaming Keyboard missing keys
- Chiclet Keyboard missing keys
- Tenkeyless (TKL) Keyboard missing keys
- Virtual Keyboard (On-screen) missing keys
- Numeric Keypad missing keys
- ISO Keyboard Layout missing keys
- ANSI Keyboard Layout missing keys
- Ortholinear Keyboard missing keys
- Bluetooth/Wireless Keyboard missing keys

**Transformative: Biological**
- An image of Lime after oxidation
- An image of breadfruit after oxidation
- An image of dragonfruit after oxidation
- An image of starfruit after oxidation
- An image of Raspberry after oxidation
- An image of Zucchini after oxidation
- An image of Pear after oxidation
- An image of passionfruit after oxidation
- An image of Blackberry after oxidation
- An image of durian after oxidation
- An image of persimmon after oxidation
- An image of Apple after oxidation
- An image of bell pepper after oxidation
- An image of olive after oxidation
- An image of Mango after oxidation
- An image of nectarine after oxidation
- An image of tomato after oxidation
- An image of quince after oxidation
- An image of coconut after oxidation
- An image of soursop after oxidation
- An image of Kiwi after oxidation
- An image of cucumber after oxidation
- An image of apricot after oxidation
- An image of Honeydew after oxidation
- An image of Peach after oxidation
- An image of pomegranate after oxidation
- An image of carrot after oxidation
- An image of fig after oxidation
- An image of Papaya after oxidation
- An image of Blueberry after oxidation
- An image of Banana after oxidation
- An image of jackfruit after oxidation
- An image of Lemon after oxidation
- An image of tamarind after oxidation
- An image of lychee after oxidation
- An image of Pineapple after oxidation
- An image of Cantaloupe after oxidation
- An image of Orange after oxidation
- An image of Rambutan after oxidation
- An image of guava after oxidation
- An image of sweet potato after oxidation
- An image of Plum after oxidation
- An image of Avocado after oxidation
- An image of Watermelon after oxidation
- An image of potato after oxidation
- An image of Grapefruit after oxidation
- An image of Grapes after oxidation
- An image of pumpkin after oxidation
- An image of Cherry after oxidation
- An image of Strawberry after oxidation
- An image of custard apple after oxidation

**Quality control**  We employ two types of quality control throughout the annotation process: an automatic check with predefined rules and a manual examination of each instance. The automatic check verifies correct MCQA format in which each question should only have one correct answer, metadata values, assesses image validity (checking the accessibility of each image) and filters out redundant images in the corpus (images that are repetitively downloaded). The manual examination

is conducted by two experts in this field, who checked the correspondence between query images and ground-truth image examples, and filtered or revised ambiguous questions and uncorrelated query image and ground-truth images.

## A.2 Human Evaluation Protocol

Three human annotators in domain conducted the human evaluation. The interface for human evaluation without RAG knowledge and with RAG knowledge are shown in Figure 6 and Figure 7.

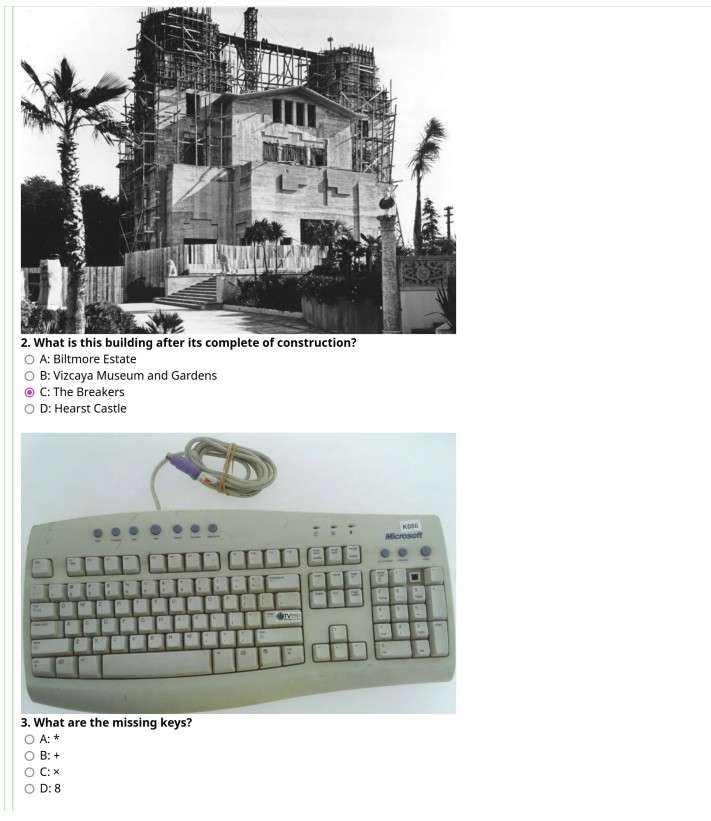

Figure 6: Human evaluation interface without RAG examples

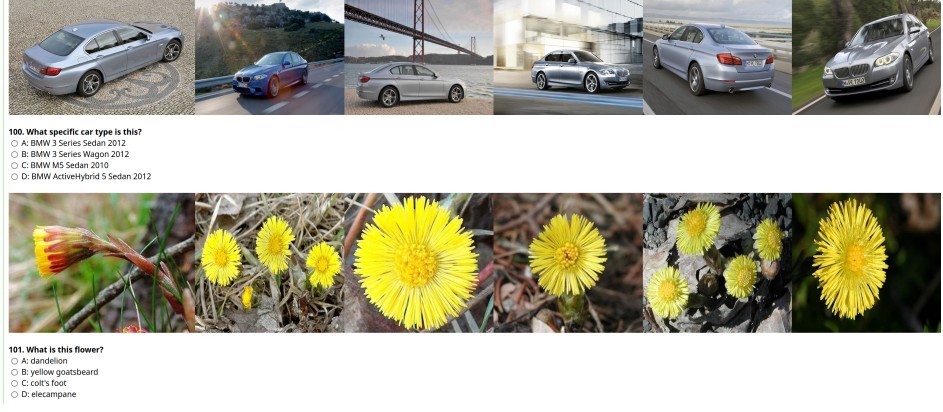

Figure 7: Human evaluation interface with ground-truth RAG examples

## B  EXPERIMENT SETTING DETAILS

### B.1  MODEL PROMPTS

Following Lu et al. (2024b) and Liu et al. (2023a) our prompt consists of four parts, the instruction, question, options, and a prefix of the answer. For images, we insert them into the text to form a coherent prompt as the image placeholder ({Image}) indicated below. The complete prompt is as follows:

---

**Model Prompts for No RAG Evaluation**

Instruction: Answer with the option's letter from the given choices directly.
{Image}
Question: {QUESTION}
Choices:
(A) {OPTION_A}
(B) {OPTION_B}
(C) {OPTION_C}
(D) {OPTION_D}
Answer:

---

**Model Prompts for RAG Evaluation**

Instruction: You will be given one question concerning several images. The first image is the input image, others are retrieved examples to help you. Answer with the option's letter from the given choices directly.
{Image}{Image}{Image}{Image}{Image}{Image}
Question: {QUESTION}
Choices:
(A) {OPTION_A}
(B) {OPTION_B}
(C) {OPTION_C}
(D) {OPTION_D}
Answer:

---

### B.2  EVALUATION TOOL

Following Lu et al. (2024b), we first use a rule-based automatic tool to extract the exact answer. First, the tool detects if a valid option index appears in the model output. If no direct answer is found, the tool matches the output to the content of each option. If there is still no match, we employ GPT-3.5-turbo to automatically extract the answer following our prompts in Table 5. If GPT-3.5-turbo finds there is still no match, we will randomly select an option as the answer.

---

**Prompt**
Please read the following example. Then extract the multiple choice letter with the answer corresponding to the choice list from the model response and type it at the end of the prompt. You should only output either A, B, C, or D.

{In-context examples}

Question: {QUESTION}
Choice List: (A) {OPTION_A} (B) {OPTION_B} (C) {OPTION_C} (D) {OPTION_D}
Model Response: {Response}
Extracted answer:

---

Table 5: Prompt template to extract multiple choice answer from model's response. {In-context examples} are in-context examples.

## C  MORE RESULTS

We present the Recall@5 scores per each scenarios on 4 multimodal retreivers as shown in Table 6 and LLaVA-Next-Interleave's accuracy score affected by these retrievers in Table 7.

| Model | Overall | Perspective | | | | Transformative | | | | Others |
|---|---|---|---|---|---|---|---|---|---|---|
| | | Angle | Partial | Scope | Occlusion | Temporal | Deformation | Incomplete | Biological | |
| MagicLens | 37.03 | 41.61 | 33.33 | 36.27 | 36.11 | 12.75 | 10.78 | 79.41 | 29.41 | 56.67 |
| E5-V | 54.92 | 49.69 | 48.78 | 61.76 | 66.67 | 38.93 | 22.55 | 73.53 | 71.57 | 82.50 |
| VISTA | 59.65 | 66.15 | 67.48 | 64.71 | 63.89 | 38.26 | 8.82 | 33.33 | 94.12 | 80.83 |
| CLIP | 60.46 | 70.19 | 54.47 | 71.57 | 73.15 | 44.30 | 31.37 | 67.65 | 40.2 | 81.67 |

Table 6: Recall@5 scores with 4 retriever models on MRAG-BENCH.

| Model | Overall | Perspective | | | | Transformative | | | | Others |
|---|---|---|---|---|---|---|---|---|---|---|
| | | Angle | Partial | Scope | Occlusion | Temporal | Deformation | Incomplete | Biological | |
| MagicLens | 35.18 | 34.78 | 29.67 | 30.39 | 34.26 | 40.94 | 36.27 | 27.45 | 49.02 | 39.17 |
| E5-V | 40.06 | 38.82 | 39.84 | 41.18 | 46.3 | 38.93 | 41.18 | 27.45 | 48.04 | 41.67 |
| VISTA | 42.42 | 40.37 | 35.77 | 40.2 | 52.78 | 45.64 | 42.16 | 36.27 | 50.98 | 48.33 |
| CLIP | 40.35 | 40.06 | 33.33 | 39.22 | 56.48 | 43.62 | 44.12 | 27.45 | 36.27 | 49.17 |

Table 7: LLaVA-Next-Interleave accuracy scores on MRAG-BENCH with 4 different retrievers.

