# OpenReview forum: "MRAG-Bench: Vision-Centric Evaluation for Retrieval-Augmented Multimodal Models"
_ICLR.cc/2025/Conference — ICLR 2025 Poster_

### Official Review · Reviewer_JirS · 2024-10-29

**Soundness:** 3
**Presentation:** 3
**Contribution:** 3
**Rating:** 6
**Confidence:** 4

**Summary:**

The authors propose a new multimodal retrieval-augmented generation benchmark, MRAG-BENCH. In this benchmark the LVLMs are given the input of query image, textual question and retrieved relevant images, and required to answer the multiple choice questions. Evaluations on MRAG-BENCH show the gap between current leading LVLMs and human beings. The authors also have an interesting findings that current open-source LVLMs lack the ability to select useful information from the retrieved images while ignore the irrelevant or misleading retrieved images.

**Strengths:**

Multimodal RAG is surely an important problem, along the improvement of multi-image understanding ability in current LVLMs. Many cases in the benchmark are close to real world scenarios. The settings of noisy retrieved information is very meaningful and necessary, since in real world multimodal RAG, errors or failures in the retrieval process are inevitable.

**Weaknesses:**

Some cases in the benchmark might be too ideal for current retrieval systems, for example ,the Angle case and the Deformation case in Figure3. It is unlikely that a multimodal retrieval system (i.e., search engine, multimodal knowledge graph) can retrieve such accurate ground truth image given the query image.

**Questions:**

Suggestions: Maybe the authors can fix the ground truth images and add retrieved images from different retrieval systems (i.e. search engine, different multimodal knowledge graph), so that this benchmark can also be used to evaluate multimodal retrieval systems.

---

> ### Author Response · Authors · 2024-11-21
> **Response to Reviewer JirS**
>
> We appreciate the construtive and insightful comments from you! We address your concerns in details below.
>
> >W1: The performance of current retrieval systems may not handle this task.
>
> Our retrieval corpus contains a diverse and highly fine-grained set of objects and images, encompassing a total of 16,130 images. This extensive collection is designed to represent a realistic retrieval corpus, effectively capturing the complexity and variability of real-world scenarios.
>
> We have demonstrated the  **sufficient retrieval accuracy** of current retriever models in our paper’s Table 6 in our Appendix C (please see line 1199). These retrieved examples provide notable benefits to all proprietary models as shown in Table 3.
>
> For instance, in the Angle scenario as you mentioned, the CLIP retriever achieves a **Recall@5 score of 70.19**, showcasing its **effectiveness in retrieving relevant visual knowledge**.
>
> Moreover, the primary focus of our benchmark is to evaluate LVLMs' abilities to **utilize** visually augmented knowledge while also establishing a baseline for the visual RAG pipeline. If certain scenarios pose challenges for the retrieval process, we see this as **an opportunity to encourage further research** and advancements in retriever models to address these limitations effectively.
>
>
> > Suggestion 1: Maybe the benchmark can also be used to evaluate multimodal retrieval systems.
>
> We thank you for your excellent suggestion! Indeed, our benchmark can also be utilized for the evaluation of multimodal retrieval systems. This further underscores the benefits of our benchmark and encourages more future follow-up research in this area.

---

### Official Review · Reviewer_tC9p · 2024-10-30

**Soundness:** 3
**Presentation:** 3
**Contribution:** 3
**Rating:** 6
**Confidence:** 4

**Summary:**

- **Introduction of MRAG-BENCH**: The paper introduces **MRAG-BENCH**, a benchmark designed for evaluating retrieval-augmented multimodal models, focusing on scenarios where visual information is more beneficial than textual data. MRAG-BENCH consists of **16,130 images** and **1,353 human-annotated multiple-choice questions** across **9 distinct scenarios**.

- **Evaluation and Findings**: The paper evaluates **10 open-source** and **4 proprietary large vision-language models (LVLMs)**, showing that models exhibit greater improvements when augmented with images compared to textual knowledge.

- **Insights and Challenges**: The findings highlight the challenges LVLMs face in effectively leveraging retrieved visual knowledge, emphasizing the need for improved utilization of visually augmented information.

**Strengths:**

### Originality
- **New Benchmark**: The introduction of MRAG-BENCH is a significant contribution, focusing on vision-centric retrieval-augmented generation, which is less explored compared to text-centric benchmarks. The paper identifies and categorizes scenarios where visual knowledge is more beneficial than textual knowledge, such as perspective and transformative changes, which is a fresh approach in the field.

### Quality
- **Comprehensive Evaluation**: The paper evaluates 10 open-source and 4 proprietary LVLMs using a large dataset of 16,130 images and 1,353 questions, providing robust and extensive analysis.
- **Detailed Analysis**: The results include fine-grained performance metrics across different scenarios, highlighting the strengths and weaknesses of various models.

### Clarity
- **Well-Structured**: The paper is well-organized, with clear sections detailing the benchmark's design, data collection, and evaluation methodology.
- **Illustrative Examples**: Figures and tables, such as the scenario distributions and qualitative examples, effectively illustrate the concepts and findings, making the paper easy to follow.

Overall, the paper makes a substantial contribution to the field of multimodal models by introducing a novel benchmark that highlights the importance of visual knowledge, backed by thorough evaluation and clear presentation.

**Weaknesses:**

- The motivation is unclear. In Figure 1, do the authors mean that retrieving from the image corpus is not difficult? While I agree that retrieving correct textual knowledge is challenging, why is retrieving from the image corpus considered easier? Additionally, I do not understand why retrieved correct textual knowledge is deemed not useful, which is confusing[^1^][1]. Please clearly state your motivation.

- How is the image knowledge database handled in the experiments? It is unclear how the authors retrieve the images.

- In my view, visual RAG is more difficult than text RAG. It would be valuable to explore CoT + Visual RAG. For example, in Figure 1, if VLLMs are unsure about the model of the car, they could consider some candidate answers and then use Visual RAG to validate and confirm the final answer. Directly retrieving images is not easy with a large image knowledge database and may sometimes hurt performance when incorrect images are selected.

- I cannot understand why GT Image RAG is better than GT Text RAG in Table 4. Can the authors provide more clarifications and illustrations?

I recommend that authors submit a link to the anonymous repo. It is very important for benchmark to release ASAP. After that, reviewers can check the data quality.

**Questions:**

Please see weakness.

---

> ### Author Response · Authors · 2024-11-21
> **Response to Reviewer tC9p**
>
> We appreciate the positive and insightful comments from you! We address your concerns in details below.
>
> > W1: Difficulty of retrieving from image corpus and usefulness and text knowledge and motivation.
>
> Please first refer to our general response for our response on motivation.
>
> We identified there are scenarios where retrieving **visual information** is either more beneficial or easier to access than textual data. Testing on our benchmark across 9 distinct scenarios, we have demonstrated the **critical role** of incorporating visual knowledge.
>
> Furthermore, we would like to clarify that retrieve **useful** image information is easier than retrieving from text corpus since we identified scenarios where image augmentation is more useful than text augmentation. Moreover, our work also motivates more research on image retrieval.
>
> For our benchmark’s scenarios, we have demonstrated the  **sufficient retrieval accuracy** of current retriever models in our paper’s Table 6 in our Appendix C (please see line 1199). For instance, in the Angle scenario, the CLIP retriever achieves a Recall@5 score of 70.19, showcasing its **effectiveness in retrieving relevant visual knowledge**. These retrieved examples provide notable benefits to all proprietary models as shown in Table 3.
>
> > W2: How is image knowledge database handled and retrieved?
>
> The collection of image knowledge databases is introduced in our paper’s 2.3 Data Collection section (line 248, 256, and 262). All of the ground-truth images from different questions formed the image knowledge base.
>
> We used CLIP as the multimodal retriever, which takes both the input image and the text question, and outputs the image candidates. We experimentally found the CLIP retriever worked the best as illustrated in our paper’s Table 6 and keep this retriever consistent throughout our experiments as mentioned in our paper’s line 415.
>
> > W3: Explore the possibility of CoT + Visual RAG.
>
> We appreciate your suggestion on this valuable direction for future work. The primary focus of this paper is to introduce a benchmark specifically designed to evaluate LVLMs' abilities to utilize visually augmented knowledge, and we will leave CoT to future work.
>
>
> > W4: Why is GT Image RAG better than GT Text RAG in Table 4?
>
> Please first refer to our general response.
> As a key contribution of this paper, our benchmark focuses on distinctive scenarios where visual knowledge is both useful and more easily accessible than textual information. For instance, consider the example in **Figure 4** of our paper. In this case, the question asks for the exact car model and make. The **GT text** includes the context of the ground truth answer (e.g., "Chevrolet Silverado 1500 Extended Cab") and additional details such as the car's year of manufacture and engine specifications, etc. Although all these information described the information of this car model, **none of them includes visual appearance** of the car. Since this example represent a difficult Angle scenario, the textual information alone is insufficient—the model **cannot recognize** the car visually due to the challenging perspective in the image. This highlights why **GT image knowledge is invaluable** in such scenarios.
>
> >Q1: Public release of benchmark.
>
> We provide an anonymous repository for evaluating the quality of our benchmark here: https://huggingface.co/datasets/mragbenchanonymous/MRAG-Bench . This repository will be made publicly available upon the acceptance of our work.

---

### Official Review · Reviewer_XA2r · 2024-11-02

**Soundness:** 2
**Presentation:** 2
**Contribution:** 2
**Rating:** 5
**Confidence:** 4

**Summary:**

This paper introduces MRAG-BENCH, a multimodal retrieval-augmented generation benchmark comprising 16,130 images and 1,353 human-annotated multiple-choice questions across 9 distinct scenarios. The authors evaluate the performance of 10 open-source and 4 proprietary large vision-language models (LVLMs) on this benchmark.

**Strengths:**

- The authors conduct an investigation into the capability of models to retrieve and utilize external visual knowledge for question answering, which is a crucial and direct exploration within the LVLM field.

- The paper introduces a comprehensive multimodal retrieval-augmented generation benchmark that provides a foundation for the community to further explore and advance in this area.

- Figure 1 is well-designed, with well-chosen examples to illustrate the benchmark's scenarios and model evaluations effectively.

**Weaknesses:**

- The data collection process is unclear. Adding a figure to illustrate the entire process could clarify the methodology, especially regarding the acquisition of ground truth (GT) text and the source of question-answer pairs.

- The experimental setup lacks sufficient detail. The authors should provide a complete subsection dedicated to describing the experimental setup, placed within Section 3, instead of scattering brief descriptions across various sections. This current layout diminishes the reliability of the experimental conclusions. For example, in Table 3 (Retrieved RAG), it’s unclear which model was used as the retriever, what the retriever’s input and output were, etc. Additional concerns regarding the experimental setup are outlined in the Questions section below.

- The analysis of the question, "How much does visual knowledge provide additional benefits over textual knowledge?" is insufficiently supported. Additional experimental analysis would strengthen the authors’ conclusions.

**Questions:**

- What is the meaning of "Unique number" in Table 2?

- How were the nine scenarios in MRAG-BENCH selected?

- Regarding the "Others" example in Figure 3, why is option A marked as the correct answer? This raises questions about the rule for obtaining question-answer pairs. How is the correct answer determined for each question?

- Could the authors clarify how CLIP is used for text retrieval? Is it a direct approach based on finding the sentence with the highest similarity to the current question, or does it involve a method similar to INFOSEEK [1], where visual entity recognition is performed first, followed by retrieval?

- Concerning Table 3:

  - If human performance on this task is low, how can the quality of GT images be assured during data collection?

  - How is "Human performance" in Table 3 measured?

  - Most open-source LVLMs exhibit performance declines when using Retrieved RAG, suggesting that retrieval effectiveness is crucial to answer quality. If retrieval consistently underperforms, is the entire process reasonable and valid?

- Concerning Table 4:

  - How were the five comparison methods implemented?

  - What is the meaning of GT Text RAG, and how is GT Text collected? Additionally, it seems counterintuitive that LLaVA-Next-Interleave performs worse with GT Text RAG. If adding GT Text RAG degrades model performance, does this suggest that the quality of GT Text might be low?

  - Why does the performance improvement for LLaVA-Next-Interleave, when provided with both GT images and textual knowledge, fall short compared to using GT images alone?

- There appears to be a minor error in Section 4.1, where it states, "LLaVA-Next-Interleave exhibited an 11.09% improvement with image knowledge over text knowledge." Shouldn't this be 11.90%?

---

> ### Author Response · Authors · 2024-11-21
> **Response to Reviewer XA2r**
>
> We sincerely thank you for your detailed review. We would first like to provide **clarifications** to all the points you raised by responding to each of your questions below.
>
> After you review our responses, we will incorporate these clarifications into our paper revision as promised. This will ensure that the revised version avoids any line number conflicts between the new and old versions, thereby improving the overall clarity and coherence of our work.
>
> > W1: Data collection of GT and source of question-answer pair is unclear.
>
> Thank you for your review. We will provide a figure in our appendix for further clarification in our revised version. The collection of image knowledge databases is introduced in our paper’s Section 2.3. Data Collection (line 248, 256, and 262). All of the ground-truth images from different questions formed the image knowledge base. As noted in line 91 of our paper, our benchmark involves **human-annotated** question answering. Therefore, the ground truth answers are derived either from class labels or human annotations, with a similar approach applied to ground truth images. More detailed responses to specific questions are provided below.
>
>
>
> > W2: Experimental Setup lacks details and missing Experimental Setup section.
>
> Thank you for your review. We clarify that we have a detailed subsection titled **Experimental Setup** in Section 3.1 (line 318 of our paper). Regarding Table 3, the retriever model used is CLIP, as mentioned in line 415. However, we agree that including this information in Table 3’s caption would enhance clarity, and we will address this in the revised version.
>
> To clarify further, the retriever’s input consists of the text question and the query image, while the output is the set of retrieved images. We appreciate your feedback and will incorporate these adjustments in our revisions.
>
> > W3: “How much does visual knowledge provide additional benefits over textual knowledge?" is insufficiently supported.  Additional experimental analysis would strengthen it.
>
> Thank you for your review. We would appreciate it if you can provide more details of what additional experiments could help strengthen this point. As demonstrated in **Table 4**and discussed in **Section 4.1** (lines 412–426) of our paper, both open-source and proprietary models show **notable performance improvements** when leveraging image knowledge than textual knowledge, whether through retrieved knowledge or ground truth knowledge.
>
>
> > Q1: The meaning of “Unique number” in Table 2.
>
> In Table 2, “Unique number” refers to a value that is distinct and not repeated within a particular context or dataset. For example, a unique number of questions means a number of questions that are not repetitive. We would clarify the wording in the revised version, thank you!
>
>
> > Q2: How were nine scenarios in MRAG-Bench selected?
>
> Thank you for your question. As discussed in our paper’s introduction (see line 92-95), we focused on utilizing visually augmented knowledge in **real-world scenarios** and divided it into two major aspects:  perspective and transformative (see paper’s section 2.1). Following their definitions, we clearly defined 9 scenarios with each scenario’s explanation detailing in our paper’s section 2.2.
>
> > Q3: Why is option A the correct answer in Figure 3’s “Others” scenario?
>
> As mentioned in our paper’s data collection section line 259-263, the “Others” scenario sourced images from GeoDE dataset (Ramaswamy et al., 2023). This dataset has class labels for different regions such as Americas, EastAsia, and we used their **class label** as the ground truth answer.
>
> > Q4: How is CLIP used for text retrieval?
>
> We used CLIP as the multimodal retriever, which takes both the input image and the text question, and outputs the image candidates. We experimentally found the CLIP retriever worked the best as illustrated in our paper’s Table 6 and keep this retriever consistent throughout our experiments as mentioned in our paper’s line 415.

---

> ### Author Response · Authors · 2024-11-21
> **Response to Reviewer XA2r (2/3)**
>
> > Q5.1: Concerning Table 3’s Human performance, if human performance is low, how is the quality of collected GT images ensured?
>
> We would first clarify that our benchmark contains some domain-specific questions that can be challenging for the average human without relying on additional tools.
>
> Human performance is measured by humans answering the question without seeing the ground truth answer and without using any additional resources such as web search.. Examples of the human evaluation interface are provided in Figure 6 and Figure 7 in Appendix A.2 of our paper. However, during the data collection process, annotators are required to **know the ground truth answers** to **ensure the quality of the ground truth data**.
>
> Additionally, human experts conducted further quality checks after the initial data collection phase. Detailed descriptions of this process can be found in Section 2.3 (lines 264–270) and Appendix A.1 (lines 1076–1082).
>
>
> > Q5.2.1: In Table 3, Since performance decline in noisy retrieved examples, retrieval effectiveness is crucial to answer quality.
>
> We agree that retrieval effectiveness is indeed crucial, so our work also points out a future research direction of improving image retrieval for VQA tasks. In addition, we clarify that, **the focus of this table** is to demonstrate the abilities of leveraging visual knowledge in LVLMs.
> As demonstrated in Table 3, most open-source models struggle to effectively leverage noisy retrieved examples, whereas proprietary models perform well with such data. This highlights a significant gap in current models (as discussed in lines 105–107 of our paper) and underscores the importance of continued research to address these limitations.
>
> > Q5.2.2 In Table 3, If retrieval consistently underperforms, is the entire process reasonable and valid?
>
> We would appreciate your clarification on the comment of “retrieval  consistently underperforms”. In fact, we have shown the **sufficient retrieval accuracy** of current retriever models in **Table 6**, and these retrieved examples provide **notable benefits** to all proprietary models.
>
> For future work, the open-source community could focus on two main areas:
> Improving Model Capabilities: Enhancing open-source models' ability to better utilize noisy, visually augmented knowledge.
> Advancing Retrieval Models: Developing more accurate retrievers, which would benefit both open-source and proprietary models.
> This dual approach can help address the challenges and improve the overall process.
>
> > Q6.1: In Table 4, How were the five comparison methods implemented?
>
> We followed the standard implementation practices for RAG systems. The code will be publicly released upon the acceptance of this paper.
>
>
> > Q6.2: In Table 4, What is the meaning of GT Text RAG, and how is GT Text collected?
>
> For retrieving textual knowledge, we clarified in the paper (lines 414–416) that CLIP is used to retrieve from a Wikipedia corpus. When retrieving ground truth (GT) textual knowledge, the input to the retriever **includes the ground truth answer** to the question, allowing it to locate the corresponding ground truth context within Wikipedia.
>
> Thank you for pointing this out. While this approach follows standard conventions, we recognize that it may not have been sufficiently clear for people from different backgrounds. To address this, we will incorporate these details into the revised version of our paper for improved clarity.

---

> > ### Author Response · Authors · 2024-11-21
> > **Response to Reviewer XA2r (3/3)**
> >
> > > Q6.3 and Q6.4 : In Table 4, It seems counterintuitive that LLaVA-Next-Interleave performs worse with GT Text RAG. and  LLaVA-Next-Interleave, when provided with both GT images and textual knowledge, fall short compared to using GT images alone?
> >
> > Once you have reviewed the response to Q6.2, we believe it also addresses the concerns raised in these related questions.
> >
> > As a key contribution of this paper, our benchmark focuses on distinctive scenarios where visual knowledge is both useful and more easily accessible than textual information. For instance, consider the example in **Figure 4** of our paper. In this case, the question asks for the exact car model and make. The **GT text** includes the context of the ground truth answer (e.g., "Chevrolet Silverado 1500 Extended Cab") and additional details such as the car's year of manufacture and engine specifications, etc. However, since this example falls under the Angle scenario, the textual information alone is insufficient—the model **cannot recognize** the car visually due to the challenging perspective in the image. This highlights why **GT image knowledge is invaluable** in such scenarios.
> >
> > Regarding concerns about the quality of GT text, as demonstrated in Table 4, GPT-4-Turbo achieves further performance improvement when combining GT text with GT image. This indicates that the **GT text quality is high**. While GT text alone may provide limited value in some cases, when paired with GT images, it becomes more meaningful and further enhances the model's performance.
> >
> >
> > > Q7: Minor error in Section 4.1, 11.09% improvement should be 11.90%?
> >
> > We appreciate your detailed observation and apologize for the typo—it should indeed be 11.90. We will correct this in the revised version of our paper. Thank you for bringing this to our attention!

---

> ### Comment · Reviewer_XA2r · 2024-12-01
>
> Thank you for your response. I want to clarify that **my primary concern regarding this paper is the lack of detailed information.** Throughout the article, details on dataset collection/processing and experimental specifics are scattered across various sections. As I read the paper, I constantly flipped back to previous sections to piece together the details I was looking for, sometimes to no avail. This current layout undermines the reliability of the experimental conclusions. Also, I agree with Reviewer "U78R" that the application scenarios of this paper are limited in real-world settings.
>
> Here are some follow-up questions, and clarifications about my concerns:
>
> My concern:
> > W2: The experimental setup lacks sufficient detail. The authors should provide a complete subsection dedicated to describing the experimental setup, placed within Section 3, instead of scattering brief descriptions across various sections.
>
> The authors' response:
>
> > We clarify that we have a detailed subsection titled Experimental Setup in Section 3.1 (line 318 of our paper).
>
> I suggest the authors provide a **complete** subsection dedicated to detailing the experimental setup. Instead of only introducing the baseline models and mentioning "follow standard MCQA evaluation setup" in line 318 of the paper. This subsection should include details about default generation hyper-parameters, the collection of human performance, the implementation of Retrieved RAG and GT RAG, and so on.
>
> **Re-Q3:** My real concern lies in the quality of the GT Example. I hope the author can provide a more detailed analysis of this example. What is the relationship between the current GT Example and the Query Image? How can the GT Example help identify if it is American?
>
> **Re-Q4:** I suggest the author try more advanced text RAG strategies, such as INFOSEEK [1]. Directly applying text RAG to the question itself greatly limits the model performance in VQA tasks. For example, in Fig. 1, the question "Which year was this building constructed？" does not contain valid information within the question itself. Therefore, using a naive Text RAG method as a baseline method seems to be inappropriate.
> [1] Can pre-trained vision and language models answer visual information-seeking questions?
>
> **Re-Q5.2:** A recall@5 of 60.46 cannot be considered "sufficient retrieval accuracy." As stated by reviewers Pyuc and JirS, the retrieval difficulty will be higher for broader application scenarios. Therefore, I remain concerned about the retrieval performance. This is a challenging issue to resolve within the entire pipeline of this paper.
>
> **Re-Q6.1:** For the authors' statement "We followed the standard implementation practices for RAG systems," please provide specific references.

---

> > ### Author Response · Authors · 2024-12-01
> >
> > Thank you for your thoughtful and engaged discussion of our work.
> >
> > We made every effort to include detailed descriptions of our experimental setup within the paper. However, due to the **page limit of 9 pages**, certain details—such as generation hyperparameters (we used default settings), the collection process for human performance data (clarified in previous responses and detailed in Appendix A.2, line 1085), and the implementation of Retrieved RAG and GT RAG (described in Appendix B, line 1138, with additional details in the anonymous code repository linked below )—were **included in the appendix** rather than the main text. These details are available for reference, and we apologize for any inconvenience caused by needing to flip between sections to piece them together. **We deeply appreciate the time and effort you have invested in reviewing our paper.**
> >
> > Regarding your observation that we mentioned "following a standard MCQA evaluation setup" (line 318), we confirm that the exact setup is described in Appendix B (starting at line 1138). We also referenced this appendix in the main text (line 333) to ensure clarity.
> >
> > >Reply of Re-Q3: Concerns about the quality of the GT Examples
> >
> > As we addressed in our responses to reviewers Pyuc and tC9p, the quality of the GT examples can be reviewed through our anonymous benchmark link: https://huggingface.co/datasets/mragbenchanonymous/MRAG-Bench. Table 3 in our paper demonstrates that **all models show significant performance improvements when using GT examples**, further underscoring their quality.
> >
> > The relationship between GT examples and query images is explained in lines 241–243, where we describe that “humanly picked five representative examples covering the diverse aspects of each class object” were chosen to assist with the query images.
> >
> > Regarding how GT examples help identify specific cases, such as whether something is “American,” we clarified in a prior response that the “Others” scenario uses images sourced from the GeoDE dataset (Ramaswamy et al., 2023). **Incorporating more images from the same class** (e.g., America) aids in identifying this aspect.
> >
> > > Reply of Re-Q4: Suggestion to try advanced text RAG strategies like INFOSEEK [1]
> >
> > We respectfully think that **no advanced text RAG strategy was employed in INFOSEEK**. **It uses standard RAG approaches**, similar to our methods. If there are specific advanced techniques we have overlooked, we would appreciate further clarification.
> > We believe INFOSEEK’s with-KB protocol using an visual recognition step is designed to parse visual information to LLMs. However, given that modern multimodal LLMs possess inherent visual perception capabilities, we believe current models and our approach more effectively interprets images and contextualizes the question.
> >
> > > Reply of Re-Q5.2: Retrieval difficulty in broader applications
> >
> > As shown in Table 3, the retrieved examples **significantly improve performance across all tested proprietary models**. While we acknowledge that retrieval difficulty increases in broader applications, our work **encourages future research** to address this challenge.  Indeed, as noted by reviewer JirS, our benchmark can also evaluate multimodal retrieval systems, making this **one of the key contributions of our work**.
> >
> > > Reply of Re-Q6.1: Standard implementation practices for RAG systems
> >
> > We followed the official implementation code for each retriever and used MagicLens (Zhang et al., 2024, ICML Oral) and E5-V (Jiang et al., 2024) as codebases for RAG evaluation. Additionally, we have provided our original code in an anonymous repository for reference: https://anonymous.4open.science/r/mragbench_retrieval-2688/README.md. We hope this demonstrates the **reliability and transparency of our experiments**.
> >
> >
> > Thank you again for your time and efforts! We remain committed to addressing any unresolved concerns or questions you may have and sincerely appreciate the time and effort you have dedicated to reviewing our paper!

---

> > > ### Author Response · Authors · 2024-12-02
> > >
> > > Dear Reviewer XA2r,
> > >
> > > As the discussion period comes to a close, we kindly urge you to review our responses. We remain fully committed to addressing any unresolved concerns or questions you may have. If all concerns have been adequately resolved in our responses, we respectfully ask you to consider reflecting this in your scores.
> > >
> > > Thank you once again for your thoughtful feedback and careful consideration.
> > >
> > > Best regards,
> > >
> > > Authors

---

### Official Review · Reviewer_Pyuc · 2024-11-03

**Soundness:** 3
**Presentation:** 3
**Contribution:** 3
**Rating:** 6
**Confidence:** 4

**Summary:**

This paper introduces a benchmark (MRAG-BENCH) designed to evaluate the performance of large vision-language models (LVLMs) in scenarios where visual information retrieval is more beneficial than textual data retrieval. First, this paper establishes the first benchmark focusing on visual information superiority over textual data. Second, this paper conducts an extensive analysis of LVLMs, and makes detailed analyses. Third, this paper points out a way that LVLMs could be evaluated and improved by emphasizing the importance of visual information in retrieval-augmented multimodal tasks.

**Strengths:**

1. The paper is well-written and easy to follow.
2. The paper offers a different visual-centric perspective and addresses a gap in the current text-centric benchmarks.
3. The comprehensive evaluation of multiple LVLMs provides valuable insights, such as how visual knowledge benefits LVLMs over textual knowledge, and the examination of retriever performance and its impact on LVLM accuracy.

**Weaknesses:**

1. The process of selecting ground-truth images and query images may introduce bias if not carefully controlled. The paper could be strengthened by discussing potential biases in the data collection process and how they were mitigated.
2. The paper focuses on a specific set of scenarios, which may not scale well to other types of visual queries or more complex tasks. For example, what if there are many objects in the question?
3. The paper could further discuss the implications of the findings, such as how they might influence the development of future LVLMs.

**Questions:**

1. How to conduct the process of human evaluation? For example, how many annotators were involved in evaluating each question, and what measures were taken to ensure consistency and accuracy in their assessments?
2. In Table 3, there are two human evaluation settings. How do you ensure that human annotators do not rely on their own knowledge when answering questions, and what steps have you taken to mitigate any potential biases in the human evaluation process
3. Whether retrieving additional visual information also enhance performance in text-intensive scenarios?
4. The paper does not report the performance of GPT-o1 on MRAG-BENCH. How does GPT-o1 perform on this benchmark?
5. In Section 4.3, the paper shows that further increasing the number of retrieved images does not improve performance. Could the authors explain the reasons behind this effect, and what insights does this provide for future research on retrieval-augmented multimodal models?
6. Will you release this benchmark?

---

> ### Author Response · Authors · 2024-11-21
> **Response to Reviewer Pyuc**
>
> We appreciate the positive and constructive comments from you, which are crucial for improving the paper!
>
> > W1: Potential bias in data collection process and mitigation.
>
> We thank you for raising this insightful concern. As discussed in the paper’s data collection section (Section 2.3), the data primarily originates from existing datasets and web-sourced images. Consequently, biases inherent in existing datasets, such as ImageNet, are inevitably present. For web-sourced data, as mentioned in line 256, we carefully curated keywords (listed in Appendix A.1, starting at line 924) and performed human filtering to ensure the quality of qualified images.
>
> One potential bias arises from the process of selecting ground truth (GT) examples. Specifically, we manually selected 5 GT images to represent the necessary visual information for each answer. This introduces a potential bias, as different individuals might select different combinations of 5 images. However, this bias is mitigated to some extent by using 5 GT images instead of just one, which already covers a broad range of possible aspects of the visual object. Furthermore, the quality of our benchmark is evidenced by its positive impact on all 14 models evaluated, as shown in Table 3 of our paper.
>
> To further mitigate this bias, a stricter control process could involve engaging more human annotators and selecting 5 GT images that reach consensus among the annotators, ensuring they are representative of the question's answer.
>
>
>
> > W2: Complex tasks when there are multiple objects in the question.
>
> We agree that when multiple visually prominent objects are present in an image, it can be challenging for retriever models to identify and retrieve relevant visual knowledge effectively. A potential solution to address this issue is to integrate an object detector model to first identify and select specific image objects. The retriever model could then focus on finding corresponding visual knowledge for the selected objects. The primary focus of our benchmark, however, is to evaluate LVLMs' abilities to utilize visually augmented knowledge, emphasizing the importance of integrating external visual information to improve performance.
>
> We believe it will be a fruitful future direction to explore multiple objects which is analogous to **multi-hop** or **multi-document** tasks in the text RAG community, where specific approaches are developed to address these challenges. For instance, such tasks often involve decomposing the question into smaller, manageable sub-questions and then retrieving relevant information from multiple documents. In our scenario, this approach could translate into searching for more images and selective feature extraction for each visual object, ensuring that the visual reasoning focuses on the appropriate elements. We will add more discussion about this future direction in our paper.
>
> > W3: Further discuss the implication of finding and how it might influence the development of LVLMs.
>
> In our analysis sections 4.2 and 4.3, we have explored several potential directions for future research. These include addressing the position bias of retrieved visual examples (line 456) and optimizing the selection of the number of necessary visual knowledge images based on the complexity of the questions (line 474). For our main findings and how explicitly to help open-source LVLMs develop better abilities in handling visual knowledge, we have the following suggestions. 1) incorporating training data utilizing visual knowledge; 2) Designing Models to Assess the Utility of Noisy Visual Information: Current LVLMs tend to treat all images as useful, which is not realistic for real-world scenarios. Models should be designed to assess whether specific visual information is helpful and selectively utilize the correct visual data. We will add more discussions to our paper.
>
>
> > Q1: How is human evaluation conducted? How to ensure consistency and accuracy of human assessments?
>
> As mentioned in our paper’s evaluation setup, the human evaluation details can be found at Appendix A.2, please see line 1085. Three human annotators in the domain conducted the human evaluation. The interface for human evaluation without RAG knowledge and with RAG knowledge are shown in Figure 6 and Figure 7.
>
> To ensure the consistency and accuracy of human assessments, we specifically provided clear rules and settings to each human annotator. As detailed in Appendix A.2, the annotators are domain experts with demonstrated expertise with similar backgrounds for knowledge consistency. To further mitigate bias and ensure consistency, we averaged the final scores from all three annotators. This approach helps reduce individual bias and improves the reliability of the overall human performance evaluations.

---

> ### Author Response · Authors · 2024-11-21
> **Continued Response to Reviewer Pyuc**
>
> > Q2: For without RAG and with RAG setting, how to ensure human annotators do not rely on their own knowledge when answering questions? Potential bias in the human evaluation process.
>
> Humans **can rely on their own knowledge** when evaluating human performance. With RAG, human performance refers to the ability to utilize combined image RAG knowledge with their inherent knowledge. This approach is **consistent** with the testing methodology applied to all models in Table 3. As observed, proprietary models tend to perform better in the no RAG setting due to the extensive knowledge encoded in their parameters.
>
> The primary focus of our benchmark, however, is to evaluate LVLMs' abilities to utilize visually augmented knowledge, emphasizing the importance of integrating external visual information to improve performance.
>
> For potential bias, we averaged the final scores from all three annotators. This approach helps reduce individual bias and improves the reliability of the overall human performance evaluations.
>
>
> > Q3: Whether retrieving additional visual information also enhances performance in text-intensive scenarios?
>
> We think this is a very interesting question! In general, text-intensive benchmarks are designed to evaluate models' abilities to utilize text-specific information, such as exact dates, years, and precise numerical records from textual corpora. Visual knowledge, on the other hand, encodes information from a different perspective, which may not be directly applicable or useful in these text-focused scenarios. We leave the study of visual retrieval augmentation for text-intensive scenarios for future work.
>
> The **primary focus** of our benchmark is to evaluate LVLMs' abilities to utilize visually augmented knowledge, emphasizing the importance of integrating external **visual information** to improve performance – a perspective that has been largely overlooked in previous studies.
>
>
> > Q4:  How does GPT-o1 perform on this benchmark?
>
> At the time we wrote this paper, GPT-o1 was just released which is 9/12/2024. However, due to rate limits of **50 queries per week** for the o1-preview model, we were unable to test it across all 1,353 questions in our benchmark.
>
> GPT-o1 primarily focuses on reasoning through complex tasks and solving more challenging problems in science, coding, and mathematics. Since it does not specifically mention enhanced visual reasoning abilities, we anticipate that its performance in our benchmark would be similar to that of GPT-4o.
>
>
> > Q5: In Section 4.3, the paper shows that further increasing the number of retrieved images does not improve performance. Could the authors explain the reasons behind this effect, and what insights does this provide for future research on retrieval-augmented multimodal models?
>
> In section 4.3’s Figure 5, we demonstrated increasing the number of retrieved images **consistently improved performance** but reached a maximum. We offered one possible explanation in the paper line 470 in section 4.3:
>
> One possible explanation could be that LLaVA-Next-Interleave may not be able to better leverage visually augmented knowledge in long context scenarios. Moreover, the complexity of questions affects the number of images needed too, one ground-truth examples sometimes help the model the most on MRAG-BENCH. We encourage the research on adaptively deciding the number of necessary images based on the complexity of questions.
>
> > Q6:  Will you release this benchmark?
>
> We provide an anonymous repository for evaluating the quality of our benchmark here: https://huggingface.co/datasets/mragbenchanonymous/MRAG-Bench . This repository will be made publicly available upon the acceptance of our work.

---

### Official Review · Reviewer_U78R · 2024-11-03

**Soundness:** 2
**Presentation:** 2
**Contribution:** 3
**Rating:** 5
**Confidence:** 4

**Summary:**

The paper presents a Multimodal Retrieval-Augmented Generation Benchmark (MRAG-Bench) to analyze and identify scenarios where visually augmented knowledge is better than textual one. MRAG-Bench comprises 16,130 images and 1,353 human-annotated multiple-choice questions, covering nine distinct scenarios. The authors analyze the performance of 10 open-source and 4 proprietary large vision-language models (LVLMs). Their results show that all the LVLMs exhibit performance improvement when augmented with visual data rather than textual. Moreover, the proprietary models can better leverage the retrieved visual information.

**Strengths:**

The paper is well-structured and accessible, presenting a benchmark for evaluating and analyzing the relevance of visually augmented samples across various scenarios. For example, it considers cases where the main object in an image is partially occluded or where the image depicts an outdated or modified version of the object. The authors conduct extensive experiments with both proprietary and open-source methods, demonstrating that visual augmentation offers greater utility than textual information for large vision-language models (LVLMs).

**Weaknesses:**

While the paper is easy to follow, the primary motivation behind the task remains unclear. In a real-world scenario, how would this pipeline function operationally? Given the findings that visually augmented samples offer significant value, an extensive repository of curated and labeled images for retrieval would be essential. However, this may be impractical due to legal and computational restrictions. Additionally, I would be interested in analyzing how the incorporation of augmented images affects response time and costs.

**Questions:**

1. Could you elaborate on your motivation for creating this benchmark? Please include examples of real-world applications where this benchmark could offer substantial value.

2. Using this benchmark in real-world applications could be challenging, as it would require a comprehensive image repository encompassing a wide range of objects globally. This need introduces various limitations, such as legal and computational restrictions and the necessity for thorough curation to prevent biased or intrusive content. Could you discuss potential strategies for making this benchmark feasible in real-world scenarios, as well as approaches to mitigate these limitations?

3. Increasing context (whether visual or textual) means that large language models (LLMs) would need to process longer input sequences, which would raise inference times. Since images are typically represented as lengthy sequences, adding more images could substantially impact both inference time and associated costs, especially in the case of proprietary models, which gain the most from visual augmentation. Could you provide a detailed analysis of the trade-offs between performance gains, increased costs, and extended inference times?

---

> ### Author Response · Authors · 2024-11-21
> **Response to Reviewer U78R**
>
> We'd like to express our sincere gratitude for your thorough review of our paper. We greatly appreciate your suggestions which are crucial in improving the quality of our paper.
>
> > W1: Extensive repositories of curated and labeled images are needed.
>
> We thank you for recognizing our **visually augmented samples offer significant value**. Notably, **our image corpus does not require labeled data**. The multimodal retriever model can search raw images and identify those that best align with the user's question and query image. Although ground-truth images can help model the most, the **primary focus** of our work is to encourage models to better **utilize noisy retrieved images**.
> As demonstrated in Table 3 of our paper, the retrieved images significantly enhance the performance of all proprietary LVLMs. For practical applications, our approach can easily integrate either publicly available web images or self-maintained raw image databases.
>
>
> > Q1: Motivation and real world application.
>
> Please first refer to our general response. We are also glad that Reviewer JirS found that **many cases in the benchmark are close to real world scenarios**.
>
> A brief example of a real-world application is the saying, "An image is worth a thousand words." Image information is inherently more diverse, capturing details, contexts, and nuances that cannot always be directly translated or recorded in a text corpus.
>
> Although the primary focus of our work is the benchmark, and we encourage future research in this direction, the **real-world applications** of our visual RAG system are diverse and impactful. These include Vehicle Identification and Insurance Claims, E-commerce and Retail, Wildlife Monitoring and Conservation, Healthcare and Medical Diagnostics etc.
>
> To illustrate one **concrete** example of the scenarios in our benchmark—perspective aspect—consider a real-world application:  When a person spots an attractive outfit on the street and wants to find and purchase it, they might take a photo in a crowded environment. This photo could capture the clothing at an awkward angle, show only a partial view, or be taken from a distance. In such cases, an LVLM can leverage its capabilities to retrieve related images from the web, enhance its visual knowledge, and provide a more accurate and useful response to the user’s query.
> The real world applications can be even broader, as in all scenarios demonstrated in our paper’s Figure 3, the models can utilize and benefit from visual image knowledge.
>
> > Q2 and W2: Legal and computation restrictions. Strategies for mitigating biased and intrusive contents in image corpus.
>
> We thank you for raising this insightful concern. As detailed in our data collection section (Section 2.3), our image corpus is sourced from both existing image datasets and web images. To ensure the quality and integrity of our benchmark, we conducted a thorough quality check (see lines 264–270) to confirm that **no intrusive content** is included. The keywords used for sourcing web images are provided in Appendix A.1. Furthermore, we adhered to all applicable licenses when legally downloading these images, and our benchmark will be released strictly for **academic purposes**.
>
> For future works aiming to construct a similar image corpus for their applications, we propose the following mitigation strategies:
> 1. Filtering Keywords: When sourcing images from the web, keywords can be filtered to exclude harmful or intrusive content.
> 2. Content Moderation by Models: During real user interactions with LVLMs, as practiced by many current systems, models can be designed to refuse to answer queries. Similarly, our approach suggests having the model refuse to retrieve potentially harmful content, even if such content bypasses the initial filtering process.
>
> It is important to note that addressing these challenges is **not the primary focus of our work**. Our benchmark is to evaluate LVLMs' abilities to utilize visually augmented knowledge while also establishing a baseline for the visual RAG pipeline. Our benchmark has demonstrated **good quality as a baseline** and we encourage more future work on large-scale adoption of our approach.

---

> ### Author Response · Authors · 2024-11-21
> **Continued Response to Reviewer U78R**
>
> >Q3 and W3: Inference cost of increasing context in multi-image models.
>
> We thank you for raising this insightful concern. There has been significant progress in multi-image models, with many strong methods excelling at handling very long image contexts. For instance, LongVILA (Xue et al., 2024) achieves an impressive 99.8% accuracy in a 6,000-frame video (more than **1 million tokens**) "needle-in-a-haystack" task.
>
> Our approach employs **only 5 additional images** (fewer than **1,000 tokens**) for visual augmentation. We believe this cost is negligible compared to the substantial performance improvements demonstrated in Table 3.
>
> Moreover, while exploring more visual contexts knowledge could be an avenue for future work, current models have already demonstrated **enough capabilities** for incorporating long visual context.
>
> Based on your question, we realize that our benchmark can also serve as a valuable testbed for long-context models, providing an additional avenue for research and evaluation. We will add this discussion in the future work section to enrich our paper.

---

> > ### Comment · Reviewer_U78R · 2024-11-26
> >
> > Thank you for the response.
> >
> > A retrieval system could be beneficial in scenarios where the image lacks clarity or contains missing details. However, its application must be approached cautiously in real-world settings. For instance, what happens if the model relies more heavily on the retrieved visual information, disregarding the original input image, particularly when the question pertains specifically to the input? Consider the car image in Figure 3 and the question, 'What happened to the car?'
> >
> > Additionally, it is important to note that in practical scenarios, instructional models are unlikely to be tasked with multiple-choice questions. Instead, their performance on open-ended questions is more relevant. This raises a critical question: could the use of retrieved images inadvertently increase the hallucination rate of these models?

---

> > > ### Author Response · Authors · 2024-11-26
> > >
> > > Thank you for the thoughtful discussion and insightful concerns.
> > >
> > > To distinguish between query input and retrieved input, the text RAG community has widely explored various established techniques. For instance, simple prompt engineering—such as using phrases like “here is the question and input” and “here are the retrieved contents”—helps models focus on the query and utilize retrieved knowledge effectively. We adopted a similar approach, as demonstrated in our evaluation prompt in Appendix B.1, line 1153.
> > >
> > > Regarding the concern that models can "heavily rely on retrieved visual information," this is precisely one of the contributions of our work. Our experiments, conducted on MRAG-Bench, highlight this **gap in current models** and underscore the motivation behind our benchmark. Importantly, MRAG-Bench is not designed to promote reliance on retrieved information, especially in noisy scenarios. Instead, it aims to encourage models to selectively and effectively utilize the retrieved information. Consequently, our work emphasizes the need for further research and advancements in this area.
> > >
> > > Finally, we opted for multiple-choice questions to **ensure deterministic evaluation**, following the methodology of well-recognized approaches in this domain, such as MathVista (Lu et al., 2024, ICLR Oral) and MMMU (Yue et al., 2024). This approach provides clarity and consistency in model evaluation.
> > >
> > > It is worth noting that open-ended questions can often be mapped to a multiple-choice format by carefully designing answer options to represent plausible outputs. While MCQs may not fully reflect all aspects of real-world scenarios, they provide a structured framework for studying critical behaviors of the models, such as their reliance on retrieved information.
> > >
> > > If the use of retrieved images increases hallucination in open-ended scenarios, as suggested, it **highlights the need for further research in this area** and **reinforces the importance of our benchmark**.

---

> > > > ### Author Response · Authors · 2024-12-02
> > > >
> > > > Dear Reviewer U78R,
> > > >
> > > > As the discussion period comes to a close, we kindly urge you to review our responses. We remain fully committed to addressing any unresolved concerns or questions you may have. If all concerns have been adequately resolved in our responses, we respectfully ask you to consider reflecting this in your scores.
> > > >
> > > > Thank you once again for your thoughtful feedback and careful consideration.
> > > >
> > > > Best regards,
> > > >
> > > > Authors

---

### Author Response · Authors · 2024-11-21
**General Response to All Reviewers**

We sincerely thank all the reviewers for their time and their constructive reviews and suggestions. We are encouraged that the reviewers find that:

 * Our work is **crucial** (Reviewer XA2r) and explores an **important** problem (Reviewer JirS), with **significant** contribution and a **fresh approach** in the field (Reviewer tC9p).
 * Our work demonstrates visual augmentation offers greater utility than textual information (Reviewer U78R), and **addresses a gap** in the current text-centric benchmarks (Reviewer Pyuc). It builds the foundation for the community to further explore and advance in this area (Reviewer XA2r).
Our benchmark is **comprehensive** (Reviewers Pyuc, XA2r, tC9p) and the evaluation of multiple LVLMs provides **valuable insights** (Reviewer Pyuc). Many cases in the benchmark are close to **real world scenarios** (Reviewer JirS). The experiments are extensive (Reviewer U78R) and offer detailed analysis (Reviewer tC9p).
* Our paper is **well-structured**, **well-written** and easy to follow (Reviewers U78R, Pyuc, tC9p), with well-chosen and **illustrative** examples to illustrate the benchmark's scenarios and model evaluations effectively (Reviewers XA2r, tC9p). The paper has **clear sections** detailing the benchmark's design, data collection, and evaluation methodology (Reviewer tC9p).


Thank all the reviewers for their valuable reviews! We believe the comments and revisions have made the paper stronger. Here is a general response to common questions and please find individual responses to your questions below.

**[Motivation]**

We would like to reiterate the motivation of this paper. As we outlined in our abstract (lines 11-15) stems from the observation that existing research primarily evaluates whether models can retrieve and utilize external **textual** knowledge. However, there are scenarios where retrieving **visual information** is either more beneficial or easier to access than textual data. Our benchmark targeted **real world scenarios** and systematically categorized into 9 distinct scenarios (see lines 91–98), each illustrated with explicit examples in Figure 3. These images encompass a diverse range of objects, and across all scenarios, we have demonstrated the critical role of incorporating visual knowledge in **real-world visual question answering**.

**[Why is image knowledge useful?]**

A simple explanation is that images capture details, contexts, and nuances that cannot always be directly translated to or recorded in  text corpora.  Besides all 9 scenarios outlined in our paper’s Figure 3 and comparisons given in Figure 1, we provide a concrete example here.
For instance, consider the example in **Figure 4** of our paper. In this case, the question asks for the exact car model and make. The **GT text** includes the context of the ground truth answer (e.g., "Chevrolet Silverado 1500 Extended Cab") and additional details such as the car's year of manufacture and engine specifications, etc. Although all these information described the information of this car model, **none of them includes visual appearance** of the car. Since this example represent a difficult Angle scenario, the textual information alone is insufficient—the model **cannot recognize** the car visually due to the challenging perspective in the image. This highlights why **GT image knowledge is invaluable** in such scenarios.

---

### Author Response · Authors · 2024-11-25
**Follow-Up on Rebuttal for "MRAG-Bench"**

Dear Reviewers,

We're following up on the rebuttal for our paper, "MRAG-Bench: Vision-Centric Evaluation for Retrieval-Augmented Multimodal Models" We appreciate the time and effort you've invested in reviewing our work.

In our rebuttal, we've thoroughly addressed the concerns and suggestions raised in the initial reviews. If you find that all your questions have been resolved, we kindly ask you to consider reflecting this in the scores. Should you have any additional questions or require further clarification, we are eager to engage in more discussion during the official ICLR discussion period.

Thank you once again for your thoughtful feedback and consideration.

Best regards,

The Authors

---

### Author Response · Authors · 2024-12-04
**Summary of Rebuttal and Discussion**

Dear SACs and ACs,

Thank you for your time and effort in handling our manuscript. We also deeply appreciate the thorough reviews provided by all reviewers.

Reviewer XA2r’s primary concerns focus on the clarity of our work and we have made efforts to address their concerns with detailed explanations and clear references to the information in our appendix.

For reviewer U78R, their concern is on further real world application of our benchmark. We explained how our benchmark provides valuable insights into real-world scenarios, as supported by Reviewer JirS.
We emphasize our primary focus has been to ensure high-quality examples and **realistic scenarios in the benchmark**, with all experiments validating these contributions. After our explanation, Reviewer U78R followed up with a question about "what if the model depends more heavily on the retrieved visual information." This question is **not a weakness** but instead aligns directly with one of the key findings of our benchmark, which underscores the need for future work to better handle noisy retrieved images. Despite addressing this concern comprehensively, Reviewer U78R became unresponsive thereafter.

For other reviewers Pyuc, tC9p, and JirS, we believe our experiments and explanations could effectively address their concerns. We emphasize improving the performance of retrieval models is one of the key areas of future work that our benchmark aims to inspire. And we have experimentally demonstrated the quality of retrieved examples with significant improved performance. **However, this aspect has been noted as a weakness by several reviewers.**

Unfortunately, despite our time and effort, the reviewers did not actively participate in the discussion throughout the rebuttal process, and therefore they did not have a deeper understanding of the contribution of this paper.

We would like to re-emphasizes the contributions of this paper:

- We introduce the first visual centric RAG benchmark that focuses on utilization of visual information, unlike previous benchmarks focusing on retrieving and utilizing external textual knowledge for question answering.

- Our benchmark consists of 16,130 images and 1,353 human-annotated multiple-choice questions across 9 distinct real-world scenarios, evaluated on 14 large vision-language models (LVLMs).

- Extensive experiments demonstrated that visual augmentation provides greater utility than textual information in our benchmark, offering **valuable insights** such as 1) how retrieved visual knowledge benefits LVLMs, 2) GPT-4o lags significantly behind human performance in visual information utilization, and 3) Open-source models are more susceptible to noisy retrieved examples than proprietary models.

- As mentioned by many reviewers, our benchmark **lays a foundation for the community** to further explore and advance in this area encouraging exploration of open questions such as reducing the effects of noisy examples, determining the optimal number of retrieved examples per task, understanding the quality and positional bias of examples, and more.

We genuinely thank all reviewers and ACs for their efforts and time in reviewing our paper, as well as their constructive suggestions that contribute to the improvement of our paper.

We sincerely hope you will take the above into consideration.

Bests,

Authors

---

### Meta-Review · Area_Chair_GB8i · 2024-12-21

**Metareview:**

The AC acknowledges the contributions of this work and significance of the problem domain, which explores the integration of retrieval systems to address challenges such as missing details or unclear input in vision-language tasks. While the approach shows promise, concerns remain about its practical implications, like,  there is a risk that the model may overly depend on retrieved visual information, potentially disregarding the original input image, which is an issue particularly problematic when the question pertains directly to the input.
Despite these limitations, the AC recognizes that the paper provides a thoughtful examination of a relevant problem and offers a foundation for further exploration. While the work has good amount of room for improvement, its contributions deserve a weak acceptance.

**Additional Comments On Reviewer Discussion:**

Good discussion and responses from the authors. Reviewer XA2r concern is about the draft's presentation clarity, and the follow up clarification from the authors addressed most of the concerns.

---

### Decision · Program_Chairs · 2025-01-22

Accept (Poster)